# AI-guided transition path sampling of lipid flip-flop and membrane nanoporation

**Matthias Post** [1] **& Gerhard Hummer** [1,2] ✉

We study lipid translocation ("flip-flop") between the leaflets of planar lipid bilayers with artificial intelligence (AI) guided transition path sampling (TPS). Rare flip-flops compete with biological machineries that actively establish asymmetric lipid compositions. By initializing molecular dynamics simulations near transition states, AI for molecular mechanism discovery (AIMMD) captures lipid flip-flop without biasing the dynamics. Four distinct mechanisms of flip-flop emerge, as encoded in neural networks trained on the fly to predict the commitment probability (or "committor") for a lipid to proceed to one or the other leaflet. Whereas coarse-grained DMPC lipids "tunnel" through the hydrophobic bilayer, unaided by water, atomistic DMPC lipids cross the membrane through spontaneously formed water nanopores. In an atomistic plasma membrane mimetic, cholesterol tunnels unaided by water, whereas PLPC lipids exploit both transient water threads and nanodroplets to cross a locally thinned membrane, as seen also in an atomistic bilayer of DSPC lipids. Remarkably, in the high (~660) dimensional feature space of the deep neural networks in AIMMD, the reaction coordinate becomes effectively linear, in line with Cover's theorem and consistent with the idea of dominant reaction tubes.

With advances in experimental techniques, the asymmetry of biological membranes has been receiving increasing attention[1,2]. The plasma membrane in particular is highly asymmetric in terms of the lipid composition of its two lipid leaflets[3–5]. While phosphatidylserine (PS) is abundant in the cytosolic inner leaf, its appearance in the outer leaflet indicates a compromised cell membrane, e.g., by a viral infection, and triggers apoptotic cell destruction[6,7]. To establish this asymmetry against an entropic driving force, elaborate ATP-driven machineries have evolved to actively translocate lipids between the leaflets[8–10] or to trap lipids on one side by covalent modifications such as glycosylation in the Golgi apparatus[11]. Without scramblases[12,13]—a class of proteins that passively redistribute lipids between leaflets—an established leaflet asymmetry tends to persist on biologically relevant timescales.

One major reason for this persistence is that spontaneous "flip-flop"[14] of lipids between the two leaflets is rare[15,16]. For flip-flop to occur, the polar or charged lipid headgroup has to pass across the apolar membrane, which is thermodynamically highly unfavorable[17]. A headgroup-dependent enthalpic cost and a tail-length-dependent

entropic cost[18] result in small rates of lipid flip-flop that decrease exponentially with bilayer thickness[19].

Lipid flip-flop rates have been measured primarily by labeling lipid headgroups, e.g., with fluorophores and spin-labels[8,19]. Label-free measurements have been limited mostly to challenging neutron-based experiments and sum-frequency vibrational spectroscopy[20]. Despite some differences between label-based and label-free measurements[18], the kinetics of flip-flop is consistently slow. Even for a zwitterionic short-chain lipid such as 1,2-dimyristoyl-sn-glycero-3-phosphocholine (DMPC), spontaneous flip-flops occur only on a minute timescale per lipid[21–23].

Molecular dynamics (MD) simulations promise a label-free view of the lipid flip-flop mechanism[24–26]. In MD, the passage of lipids flipping their membrane orientation can be studied in full microscopic spatio-temporal detail. For instance, previous numerical studies noticed a connection of flip-flop to the formation of transient water pores[24]. Artificially forcing single lipids to move into the bilayer results in water defects, which then may span the whole bilayer[27]. Conversely, creating

[1]Department of Theoretical Biophysics, Max Planck Institute of Biophysics, Frankfurt am Main, Germany. [2]Institute of Biophysics, Goethe University Frankfurt, Frankfurt am Main, Germany. ✉e-mail: gerhard.hummer@biophys.mpg.de

pores in a membrane (e.g., in case of ionic charge imbalance[28–31], via electroporation[32–34], or lateral/osmotic stress[35]) allows lipids to cross between leaflets by diffusing along the membrane edge lining the pore. The free energy cost for pore formation is known to increase with membrane thickness,[36,37] with a trade-off between enthalpy and entropy.[24]

Spontaneous lipid flip-flop has thus at least two conceivable, distinct reaction channels—even if one ignores the assistance by membrane protein scramblases or other membrane insertions. In the "tunneling" pathway (denoted $\Pi_T$ in Fig. 1a), the phospholipid flips in isolation, with its headgroup passing through the bilayer and its acyl chains reorienting in the membrane. In the pore pathway (denoted $\Pi_P$), a water-filled pore transiently opens in the membrane and one or several lipids then traverse across the pore-lining membrane edge before the pore closes again. The vertical position $z$ of the headgroup is a natural "reaction coordinate" for transversal displacement. As coordinate for the presence and size of a possibly associated pore, we use $\xi_P$ by Hub[36], which accounts for the occupancy of polar atoms within the midplane[38,39] and their mean (lateral/axial) distance to the nucleation center[40].

To resolve the dominant mechanism among multiple pre-identified choices—here direct versus pore-mediated flip-flop—one could try to calculate and compare the respective transition rates. However, this often proves challenging, in particular for a process occurring on the minute timescale. Two common strategies to overcome this difficulty are coarse-grained models[41,42], i.e., representing the lipids and solvent by larger beads, and including a steering bias, e.g., by umbrella sampling[37]. Coarse-graining tends to result in much faster kinetics and also in less stable water pores due to the simplistic interaction potential and entropy loss[43]. Conversely, steering may result in inadequate estimates, in particular if degrees of freedom orthogonal to the chosen bias are relevant for the process.

Here, we use the recently developed "Artificial Intelligence for Molecular Mechanism Discovery" (AIMMD)[44]. In AIMMD, we apply transition path sampling (TPS)[45] to harvest reactive trajectories without the application of bias forces or the choice of predefined collective variables or reaction coordinates. From TPS, we learn the commitment probability (or, in short, committor)[46,47] on-the-fly, encoded in a deep neural network. As the probability to proceed to the product state for a given starting configuration, the committor pinpoints important microscopic features describing the reaction mechanism. The features used as inputs for the neural net include in particular the positions of neighboring lipids in a symmetry invariant form (i.e., their transversal distance between heads, $\Delta z^{PO4}$, and the distances between individual pairs of atoms, $\Delta r^{all}$, as depicted in Fig. 1b). In addition, we include reporters on the nearby hydration, with the water-pore coordinate $\xi_P$ of Hub[36] as a primary input. From the influence of the features on network accuracy we then deduce the importance of factors ranging from lipid orientation to water nanoporation. For the latter, we benefit from extensive earlier studies[36,40,48,49].

While very early pioneering work studying lipid flip-flop via TPS only used coarse-grained models[50,51], AI-guidance in AIMMD allows us to study the molecular mechanism in full detail by sampling hundreds of lipid flip-flop events in atomistic MD simulations. We apply this general framework to neat DMPC lipid bilayers, as a single-species model used extensively in systematic studies of various membrane properties[22,52–54], including lipid flip-flop. We compare results for atomistic MD simulations with those obtained using Martini coarse-graining.

By seeding the AIMMD simulations with initial paths in the two extreme pathways $\Pi_T$ and $\Pi_P$, we establish the relaxation of the TPS to the dominant mechanism. In this way, we show that DMPC lipids prefer tunneling in the Martini model and pore-formation in the all-atom MD model. Beyond the mechanism of lipid flip-flop, AIMMD also discovers the mechanism for the spontaneous formation of a membrane pore in a DMPC lipid bilayer, as a combination of pore size ($\xi_P$) and vertical lipid displacement ($z_1$). For thicker bilayers formed by long-tailed 1,2-distearoyl-sn-glycero-3-phosphocholine (DSPC) lipids, lipid translocation is catalyzed by narrow and transient water threads across a locally thinned, hour-glass-shaped membrane. We confirm that "dry" tunneling predominates for cholesterol flip-flop in MD simulations of a plasma membrane mimetic[26] with leaflet asymmetry, whereas PC lipids cross between leaflets both along transient water nanowires and solvated by small water nanodroplets.

## Results
### Martini DMPC lipids prefer tunneling mechanism
We start our investigation with a coarse-grained DMPC lipid model, referring to Methods for detailed descriptions of the MD simulations and TPS setup. The flip-flop transition of an individual lipid (the "probe lipid") between the lower leaflet (state $\mathcal{L}$) and the upper leaflet (state $\mathcal{U}$) is tracked by monitoring its transversal displacement $z$ from the center of the lipid bilayer. We also measure how the hydration state of the lipid bilayer around the probe lipid changes over the course of the Monte Carlo (MC) chain of transition paths. A TPS MC step here corresponds to one two-way trajectory shooting attempt. Figure 2a shows the (time) averaged pore reaction coordinate, $\hat{\xi}_P$, as a function of the MC step $n$ in TPS, where values $\hat{\xi}_P \gtrsim 1$ indicate the presence of a membrane-spanning water pore. We track each of the samplers starting from $\Pi_T$ (red) and $\Pi_P$ (blue) individually (faint) as well as their mean (solid).

In $\Pi_T$, we initiate the TPS MC chains with an intact flat DMPC double-layer without pore, $\hat{\xi}_P \approx 0.1$. Over the course of the MC chain, $\hat{\xi}_P$ continues to fluctuate around that value. Thus, the transition mechanism remains in $\Pi_T$, i.e., without the utilization of water pores. By contrast, when starting from $\Pi_P$, the initial, artificially large pore rapidly shrinks from $\hat{\xi}_P > 1$ towards $\approx 0.5$. For the first few hundred MC steps, $\hat{\xi}_P$ does not further drop, i.e., the water pore is not fully closed. In this intermediate phase of TPS, the probe lipid is still connected to neighboring water beads, e.g., via a connecting water thread

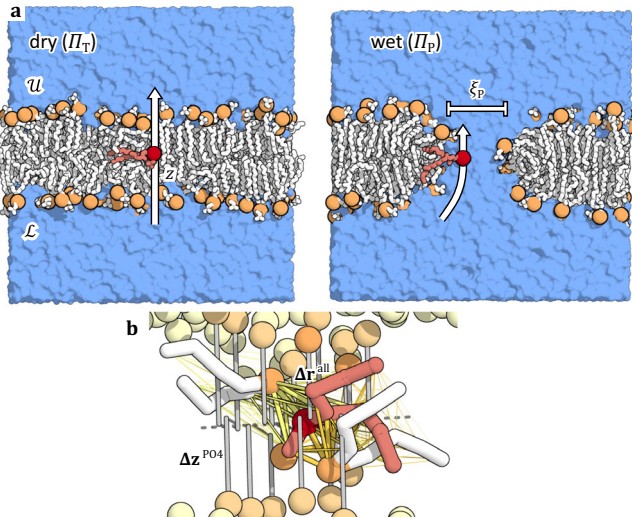

**Fig. 1 | Schematic of lipid flip-flop simulations. a** Sketch of the simulation set up and the two initial transition pathways. (Left) The "probe" lipid (red) is pulled from the lower leaflet (state $\mathcal{L}$) to the upper leaflet (state $\mathcal{U}$) to produce a "dry" initial pathway $\Pi_T$ "tunneling" through the bilayer. (Right) Alternatively, the probe lipid (red) moves along the edge of a pre-established water nanopore in the "wet" pore pathway $\Pi_P$. Lipids are shown as sticks, phosphorus as orange ball, and water as surface. **b** Individual atom distances $\Delta r^{all}$ to neighboring lipids as the most predictive input features of the neural networks describing the reaction mechanism in terms of the committor (yellow), together with vertical displacements of lipid phosphate groups ($\Delta z^{PO4}$).

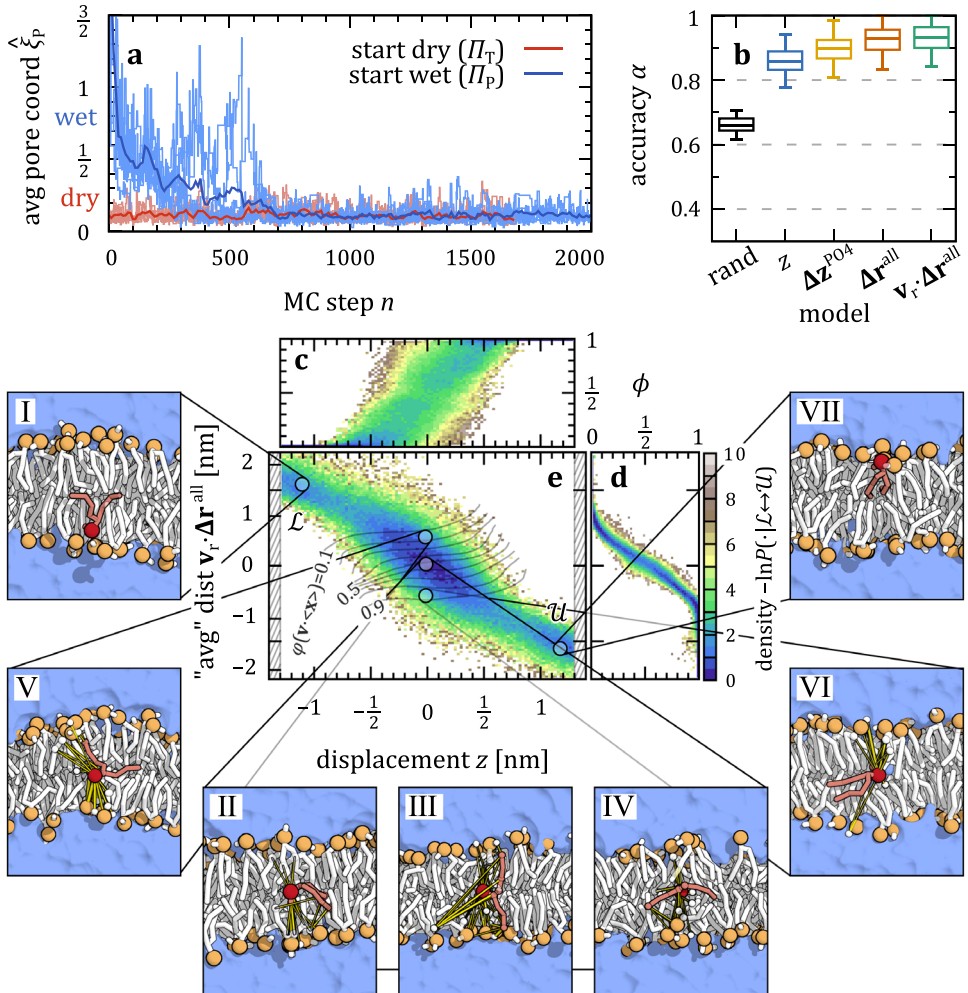

**Fig. 2 | Coarse-grained DMPC lipids tunnel through membrane. a** Time-average of pore reaction coordinate, $\xi_P$, evolving during the TPS MC chain. We compare samplers starting from an intact membrane (dry path $\Pi_T$, red) and from a formed pore (wet path $\Pi_P$, blue). Dark colors show a sample average, smoothed over 10 TPS MC steps. **b** Accuracy of committor models comparing different input features to random committor assignments ("rand"). Boxes show median and 25/75 th percentile of 1000 bootstraps drawn from a total of 10000 MC steps; whiskers show 2.5/97.5 th percentile. **c, d** Distribution of committor estimates for a given feature, comparing transversal displacement $z$ (**c**) with a linear combination of bead-to-bead distances to neighboring lipids, $\Delta\mathbf{r}^{all} \cdot \mathbf{v}_r$ (**d**). **e** Projection of the TPE onto $z$ and $\Delta\mathbf{r}^{all} \cdot \mathbf{v}_r$. Gray iso-lines show the committor averaged over 5000 nearest-neighbors (0.6% of all data). Representative configurations (dots) are shown in the seven side-panels I–VII. Nearest neighbors are colored in white with increasing intensity, and distances in yellow.

(see Supplementary Fig. 1a for an exemplary transition). The connection breaks, though, as the probe lipid reaches the other side, flushing out all water beads of the membrane (see also Supplementary Fig. 1b). Notably, in this intermediate, unstable period, the transition times to move from $\mathcal{L}$ to $\mathcal{U}$ (or vice versa) are the smallest, even compared to $\Pi_T$ (see Supplementary Fig. 1c). After about $n \approx 500$ MC steps on average, we have a behavior similar to $\Pi_T$, and thus, all water beads are flushed out and the pore is closed completely during the remaining transitions. There are only rare occasions of single water beads penetrating the membrane, even while the probe lipid is situated in the mid-plane (see also Supplementary Fig. 1b).

By initializing the MC samplers from the two competing mechanisms $\Pi_T$ and $\Pi_P$ and observing that all TP samplers converged to $\Pi_T$, we clearly see that Martini DMPC lipids prefer to flip-flop without utilizing transient water pores. To further explain how they instead tunnel through the bilayer, we study the importance of individual microscopic features $\mathbf{x}$ describing the committor $\phi(\mathbf{x})$. To that end, we train on the $\Pi_T$ data, evaluate a variety of neural network models of $\phi$ and measure their respective TP prediction accuracy.

Figure 2b compares how the use of different input features affects the accuracy $\alpha$ of the model; see "Methods" for its definition. While its

transversal displacement $z$ (blue) already does a reasonable job predicting TPs, we find that a full description of the tunnel transition mechanism requires adding direct information about the neighboring lipids, like the vertical position $\Delta\mathbf{z}^{PO4}$ of their PO4 beads, or better via the relative distances, $\Delta\mathbf{r}^{all}$ (red), of beads of the lipid neighbor network, then giving close-to-optimal prediction accuracy. We refer to Methods and Supplementary Table 1 for details on the network architectures.

When we now train a network model on all these features, and try to understand how they are encoded into $\phi(\mathbf{x})$, we find that the direction $\mathbf{v}(\phi) = \langle\nabla\phi\rangle_\phi / |\langle\nabla\phi\rangle_\phi|$ of the reactive flux averaged on iso-surfaces of $\phi$ hardly changes with $\phi$ (see Supplementary Fig. 2a). This implies a simple shape of the committor, $\phi \approx \varphi(\mathbf{x} \cdot \mathbf{v})$, i.e., a quasi-linear model of the input features. The flux direction $\mathbf{v}$ in feature space emphasizes the tilt angle[31,55,56] of the probe lipid even more than its vertical position $z$. Yet, most of the weight is found in the distances of its head to neighboring lipids (denoted as a weight vector $\mathbf{v}_r$; see also Supplementary Fig. 2b). A simple model using the linear projection $\mathbf{x} \cdot \mathbf{v}$ as input resolves and reproduces the committor (Fig. 2b and Supplementary Fig. 2c, d). A possible interpretation of this weighted average of neighbor distances as reaction coordinate may be that the

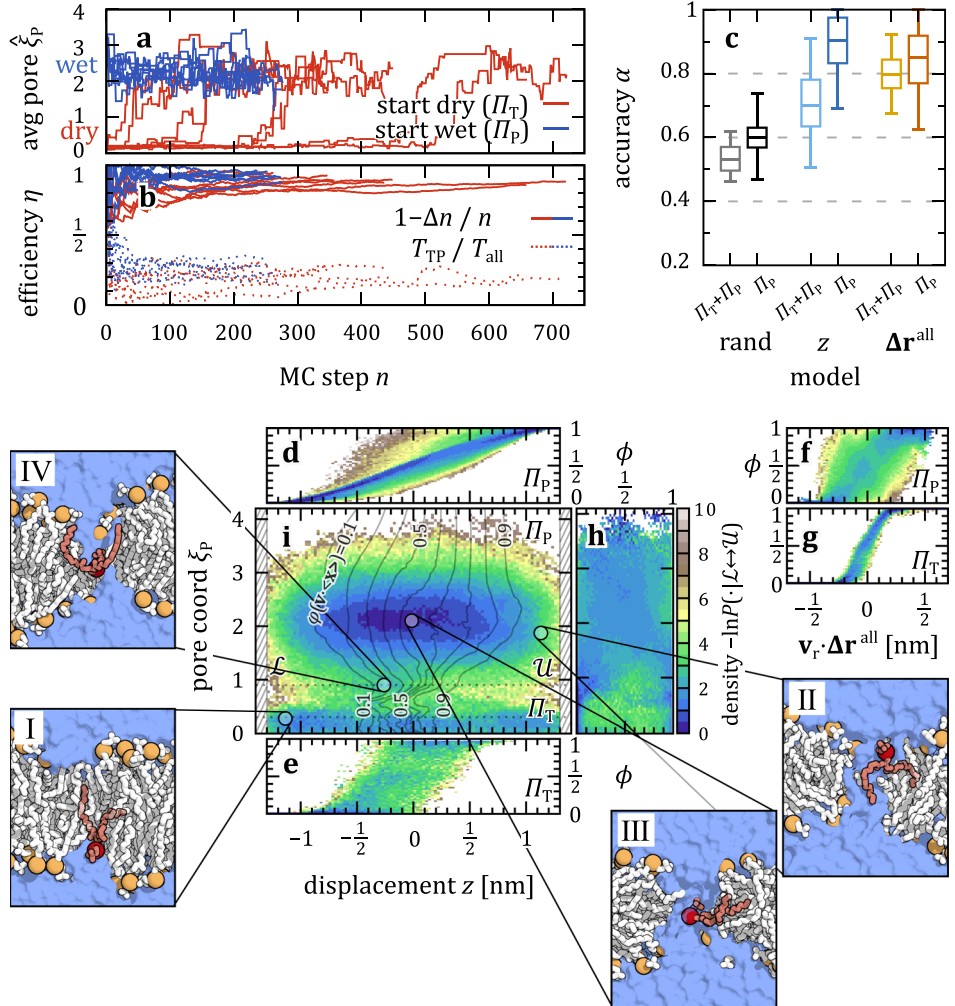

**Fig. 3 | Atomistic DMPC lipids flip bilayers through water-filled nanopores.**
**a** Time-average of $\xi_P$ during the MC chains, comparing sampler starting from the tunnel mechanism, (dry, $\Pi_T$, red), with those starting with a pore (wet, $\Pi_P$, blue).
**b** Efficiency $\eta$ measured by difference of expected and generated TPs, $\Delta n = n_{exp} - n_{gen}$, and by the simulation time $T_{TP}$ of new transition events compared to the total simulation time $T_{all}$. **c** Accuracy of committor models correlating the vertical lipid displacement $z$ to bead-to-bead distances $\Delta \mathbf{r}^{all}$, training on all data (bright), compared to only $\Pi_P$ (dark). Boxes show median and 25/75th percentile of 800 bootstraps drawn from a total of 3500 (1418 in $\Pi_P$) MC steps; whiskers show 2.5/97.5th percentile. Random committor assignments: rand. **d–h** Distribution of committor estimates for a given feature, comparing $z$ (**d**, **e**) and a linear combination of distances (**f**, **g**), $\Delta \mathbf{r}^{all} \cdot \mathbf{v}_r$, stratified to the $\Pi_P$ (**d**, **f**) and $\Pi_T$ (**e**, **g**) data, and the pore coordinate $\xi_P$ (**h**). **i** Projection of the TPE onto $z$ and $\xi_P$. Black iso-lines show the committor averaged over 5000 nearest-neighbors (0.6% of all data). Representative configurations are shown in the four side-panels I–IV.

network learned how to better identify the geometry and center of the membrane. But the linearity of this model also implies that there is no particular sequence of events (no specific conformational change of the lipid and its neighbors) resulting in flip-flop.

It is instructive to compare the now identified important projected neighbor distance, $\Delta \mathbf{r}^{all} \cdot \mathbf{v}_r$, with the probe lipid's vertical displacement $z$ based on their committor estimates (Fig. 2c, d) and the transition path ensemble (TPE) projected onto these features (Fig. 2e). The flip-flop starts by inserting the lipid tail-first from one leaflet (state $\mathscr{L}$ in panel I) into the bilayer. It then tilts into the cavity inside the midplane, where a variety of distinct conformations with the same insertion depth $z$ have the same commitment probability (Fig. 2c): e.g., conformations with joined tails (panel II) and with split tails either parallel (panel III) or perpendicular (panel IV) are projected to roughly the same spot, balancing the distances to the two leaflets. The projection $\Delta \mathbf{r}^{all} \cdot \mathbf{v}_r$ of distances onto $\mathbf{v}_r$ resolves the configurations $\mathbf{x}$ according to their committor values $\phi$ (Fig. 2d, and the (gray) iso-lines of $\varphi\left(\langle \mathbf{x} \rangle_{z, \Delta \mathbf{r}^{all} \cdot \mathbf{v}_r} \cdot \mathbf{v}\right)$ in Fig. 2e). Note, though, that close to the state boundaries, defining the $\phi = 0$ and 1 iso-surfaces, this simple linear

model has to fail. For a given configuration, the linear projection $\Delta \mathbf{r}^{all} \cdot \mathbf{v}_r$ identifies the features that commit it to one or the other leaflet as the probe head and leaflet phosphates approaching each other (Panel V and panel VI). At the end, the lipid is pushed out straight (state $\mathscr{U}$ in panel VII). See also Supplementary Fig. 3 for time-traces of exemplary TPs.

## Charmm36 DMPC lipids utilize transient water pores

We repeat the same procedure with an all-atom representation of the DMPC lipids; see "Methods". We again classify the overall mechanism of transition by means of the pore defect $\hat{\xi}_P$, as shown in Fig. 3a (top). The samplers prepared initially in $\Pi_P$ (blue) all stay in the pore state, $\hat{\xi}_P > 2$. By contrast, the samplers initiated in $\Pi_T$, i.e., with an intact membrane (red), all start with $\hat{\xi}_P$ well below 1. Still, each individual sampler eventually switches to $\hat{\xi}_P \approx 2$ as the TPS MC chain progresses. As TPS progressed, lipid flipping thus triggered the formation of water nanopores across the bilayer, which were then kept intact throughout the remaining MC chain. While the probe is able to drag a few water molecules from the get-go (see, e.g., the average neighboring water in

Supplementary Fig. 4a, b for exemplary trajectories), a switch to a fully connected water chain ($\hat{\xi}_P \approx 1$) is relatively abrupt, meaning the pore is quickly filled during only a few MC steps. The pore then finally relaxes to about twice its initial size, in accordance with ref. 36 (see also Supplementary Fig. 5 for exemplary TPs). The open nanopores persist for about 0.4 μs on average, far beyond the ≈ 25 ns long flip-flop events (Supplementary Fig. 6a, b). AIMMD thus shows that in the atomistic model, DMPC lipid flip-flop is associated with the formation of water nanopores across which the lipids then traverse between the leaflets.

We can quantify the efficiency of the AI-guided path sampling in AIMMD by comparing the number of actually generated reactive trajectories ($n_{gen}$) to those expected for the estimated commitment probabilities ($n_{exp}$, given by the cumulative sum over the estimated TP probability $P(\text{TP}, |, \mathbf{x}) = 2\phi(\mathbf{x})[1 - \phi(\mathbf{x})]$ [57]. Figure 3b (solid lines) shows $\eta_{\Delta n} = 1 - \Delta n / n$ as a measure of efficiency, with $\Delta n = n_{exp} - n_{gen}$ and $n$ the number of MC steps. Convergence to $\eta_{\Delta n} \approx 0.88$ indicates a good network model of the committor $\phi$ also for the atomistic MD simulations.

We alternatively measure the efficiency by $\eta_T = T_{TP}/T_{all}$ comparing the aggregate time $T_{TP}$ of newly accepted transition paths entering the TPS Markov chain to the total simulation time $T_{all}$ (Fig. 3b, dashed lines). With ∼ 18% of MC steps resulting in accepted TPs (16% of $\Pi_T$, 23% of $\Pi_P$) of a combined time $T_{TP} \approx 12$ μs, we achieve an efficiency of $\eta_T \approx 0.25$, i.e., a quarter of the time goes to simulating new, accepted transition paths.

The analyses of the TPS data and the committor model trained on the shooting results now depend on whether or not we include the large portion of initial $\Pi_T$ transitions. If we do, see Fig. 3c, we see that the committor prediction using all neighbor distances $\Delta \mathbf{r}^{all}$ (yellow) again outperforms a simple model using $z$ alone (light blue). With the smaller sample size, the model accuracy is worse than in the coarse-grained case (see Supplementary Fig. 7a for an error analysis), but we again see the importance of the precise relative position to the probe's neighbors for a successful transition.

If we, however, consider only the data from the dominant path-type $\Pi_P$ to which all TPS walkers relax eventually, the transversal displacement $z$ (dark blue) suffices to describe the transition mechanism. There is no improvement by using more features, like the distances to neighbors, $\Delta \mathbf{r}^{all}$ (red). While we expect that the AIMMD efficiency would slightly improve when continued sampling, our collected data (1419 $\Pi_P$ paths for in total 3500 shooting points (SPs)) is convincing enough to confirm the diffusion along $z$ via $\Pi_P$. We also refer to Supplementary Fig. 7b, c for cross-validation of these models.

We again find that the training process resulted in learning a linear combination of the input features and a uni-directional reactive flux (see Supplementary Fig. 8). In Fig. 3d–i we break down its main contributors: the displacement $z$ (d,e; with more weight compared to Martini), and the distance average $\Delta \mathbf{r}^{all} \cdot \mathbf{v}_r$ (f,g; with very similar weight); and compare it to the pore-defining reaction coordinate $\hat{\xi}_P$ (h). To no surprise, we see differences in prediction accuracy of these features depending if we limit the analysis to either the $\Pi_T$ or $\Pi_P$ data. That is, in case of $z$, we see a relatively sharp distribution of $\phi$ when traversing along $\Pi_P$ (Fig. 3d), again indicating that $z$ is capable to describe the diffusion. For the $\Pi_T$ mechanism, however, $\phi(z)$ broadens (Fig. 3g), which means $z$ is a poor descriptor of the pore-less flip-flop, in line with our Martini result. Conversely, if we look at the weighted average of distance between atoms of probe and neighboring lipids, $\Delta \mathbf{r}^{all} \cdot \mathbf{v}_r$, we see a broad distribution corresponding now to $\Pi_P$ (Fig. 3f), and a sharp distribution in $\Pi_T$ (Fig. 3g). This tells us that $\Delta \mathbf{r}^{all} \cdot \mathbf{v}_r$ takes a similar role as in the Martini case, describing the pore-less tunneling via $\Pi_T$ and becoming obsolete after TPS has converged to $\Pi_P$, where $z$ alone suffices. Including $\Delta \mathbf{r}^{all}$ again improves the localization of the effective membrane center of an intact membrane, but not when situated in a pore.

The state $\xi_P$ of the water pore, in contrast, is always a poor predictor and is thus not deemed important by the model. Figure 3e shows the TPS data projected onto the displacement $z$ and pore shape $\xi_P$. The bottom region, $\xi_P < 1$, represents early trajectories traversing from $\mathscr{L}$ to $\mathscr{U}$ via $\Pi_T$ (see also panel I). Conversely, the upper $\xi_P > 1$ region shows the trajectories starting and ending with a formed open pore, with fluctuations around $\xi_P \approx 2$ due to pore expansion and contraction (panel II and III). The transition from $\Pi_T$ to $\Pi_P$ paths in the TPS MC chain itself appears to be a rare event, associated with the nucleation of a water nanopore in a single trajectory, as reflected in a step in $\xi_P$ (panel IV). Here, the probe's head within the membrane attracts the surrounding water to seed and eventually form a percolating water thread. The iso-lines of $\phi$ projected onto $z$ and $\xi_P$ expand from the narrow $\Pi_T$ to a broader and less committed behavior along $\Pi_P$, but stay roughly parallel to $\xi_P$ otherwise (see also Supplementary Fig. 7d for a study of the midplane-symmetry).

Without imposing a sealed membrane in the initial and final state, the network model thus did not need to learn the actual transition mechanism of flip-flop, but only the intermediate, diffusive step (along $z$), before and after the formation and closing of the water pore ($\xi_P$). This is because we defined the states $\mathscr{U}$ and $\mathscr{L}$ only via the displacement $z$ such that we observe pore nucleation only during the initial equilibration phase of TPS.

## Pore nucleation precedes flip-flops

Therefore, we now aim to capture the nanopore nucleation step prior to the lipid traversal for a full description of the flip-flop process. So far, nucleation of water pores was achieved by merely shooting close to the transition state, hinting at the importance of flip-flopping lipids as seeds for the formation of transient water pores. These nucleation events are now used as TPS starting points, and evaluated via the pore reaction coordinate $\xi_P$ to define the flat and porous membrane. We refer to Methods for simulation details.

We see in Fig. 4a that AIMMD is capable of efficiently sampling also pore nucleation. With efficiencies of $\eta_{\Delta n} \approx 0.95$ and $\eta_T \approx 0.24$, we have a total of 256 distinct TPs to analyze the mechanism of pore nucleation. Since $\xi_P$ is treating all lipids as a group, instead of having one tagged lipid probe, we look at the behavior of each lipid, sorted, e.g., by their distance $z_i$ from the midplane.

To elucidate the mechanism of pore nucleation, we again compare different features as inputs for the committor network model, Fig. 4b. With further details in Methods, we compare using only $\xi_P$ with using $z_1$ as inputs. Inspired by the work of ref. 40, we also use the largest depletion of four P and N atoms from the nucleation center, $\Delta z_{NP1-4}^{max}$. While there is a hint of better accuracies $\alpha$ using the latter, both $\xi_P$ and $\Delta z_{NP1-4}^{max}$ do a decent job in predicting TPs. There is no major improvement in $\alpha$ by using more input features, which confirms their suitability as reaction coordinates. The somewhat lower accuracy of the committor models for the atomistic DMPC model ($\alpha$ between 0.8 and 0.9 in Fig. 3c) compared to the Martini DMPC model ($\alpha \approx 0.9$ in Fig. 2b) is likely due to a combination of fewer training data and thus some not fully resolved atomistic details (with pore and tunnel mechanism).

To study the connection of pore nucleation to lipid flip-flop, we start by simply counting all observed translocation events. During most nucleation transitions, no lipids flip. We only observe 10 distinct flip flop events, in ≈ 3% of the TPs. See Supplementary Fig. 9a, b for an example of these transitions. In all of these cases, the flip-flop is preceded with bulging of the membrane and then water forming a percolating thread between the leaflets (see Supplementary Fig. 9c for evaluation of $z_1$ compared to the pore in these cases). So, while our SP selection in the previous sampling of lipid flip-flop inevitably resulted in the nucleation of water-pores, we can rule out lipid flip-flop as a main, native trigger of pore nucleation. Instead, the flip-flop happens

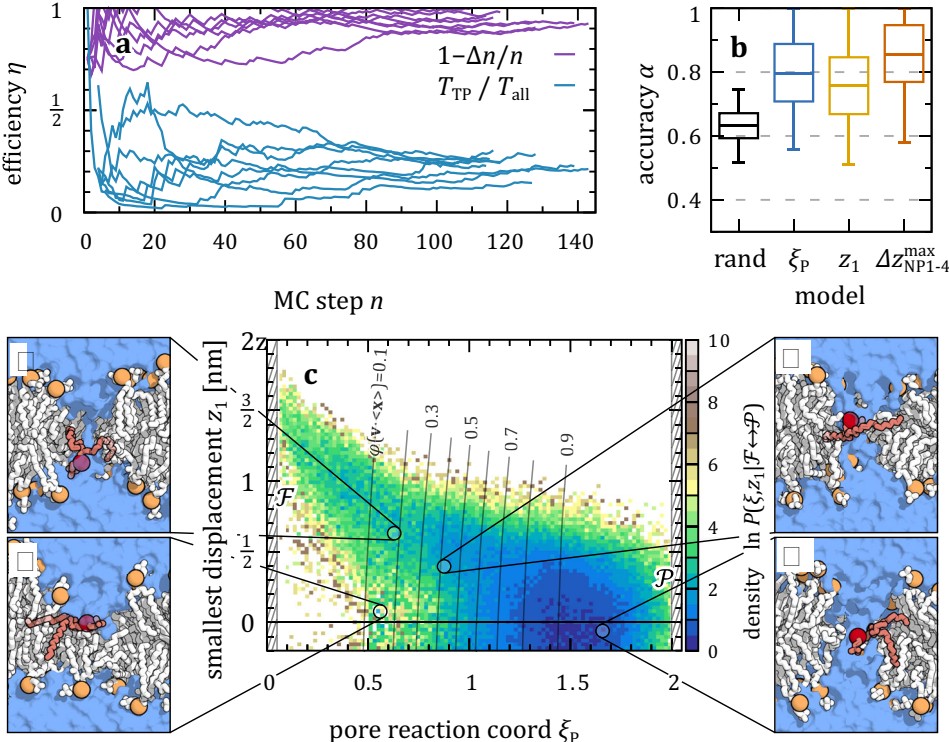

**Fig. 4 | TPS of water nanopore nucleation in membranes of Charmm36 DMPC lipids. a** Efficiency of AIMMD measured by the difference of expected and generated TPs, $\Delta n = n_{exp} - n_{gen}$, and the ratio between simulation time of new transitions, $T_{TP}$, and total simulation time, $T_{all}$. **b** Accuracy of committor models, correlating pore reaction coordinate $\xi_P$ to closest lipid displacement $z_1$ and the depletion of P and N atoms $\Delta z_{NP1-4}^{max}$. Boxes show median and 25/75th percentile of 800 bootstraps drawn from a total of 1000 MC steps; whiskers show 2.5/97.5th percentile. **c** Pore nucleation mechanism. The center plot shows the TPE for nucleation of a pore (state $\mathscr{P}$) starting from a flat membrane (state $\mathscr{F}$), projected onto the plane spanned by $\xi_P$ and $z_1$. Side panels show representative structures.

at a later stage, with the spontaneously formed nanopores staying open for about $0.4\ \mu s$ on average, with typically about 15 flip-flop events (Supplementary Fig. 6a).

By sampling the nucleation process, still, some lipids have to migrate towards the bilayer midplane. The TPE in terms of $\xi_P$ and $z_1$, see Fig. 4c, shows how (at least) the lipid closest to the nucleation center migrates into the pore. We see that starting from a flat surface, state $\mathscr{F}$, the membrane starts to bulge and thin locally, thus also bringing $z_1$ closer to zero. As $\xi_P$ reaches 0.5, a water connection to the other side forms. The insertion of water molecules is then followed by the polar lipid heads, see Supplementary Fig. 9d, in accordance with refs. 35,40,58. A pore is formed for $\xi_P > 1$, which then has to stretch to a slightly expanded shape to reach state $\mathscr{P}$.

### Transient water threads and local membrane thinning as third mechanism for flip-flop through thick membranes

For thick membranes formed by long-tailed DSPC lipids, yet another mechanism emerges: flip-flop mediated by transient and narrow water threads associated with local membrane thinning. We initiate AIMMD simulations of atomistic DSPC lipid bilayers from $\Pi_T$ and $\Pi_P$ initial pathways (Supplementary Fig. 10). In the $\Pi_P$ sampler, the water pores quickly become narrow, and collapse almost immediately after completed flip-flop. This collapse is consistent with water nanopores being disfavored in thick bilayers[36,37,59]. By contrast, the initial "dry" flip-flop in the $\Pi_T$ samplers quickly changes to incorporate water, where the probe head drags individual water molecules as a single shell to the other side. In one case, the flip-flop mechanism transitions to a narrow pore that then persists. Visual inspection shows that the process is initialized by bulging, resulting in local thinning of the membrane, after which a transient water thread[48] forms (see the examples in

Supplementary Fig. 11). The flipping lipid then connects the two DSPC leaflets, which adopt a shape resembling a conic intersection. However, more extensive TPS would be needed for a full quantification of the reactive flux carried by this third reaction channel, intermediate between the "wet" and the "dry" pathways with and without fully formed water nanopores.

### Dry and wet flip-flop in plasma membrane

To study lipid flip-flop in a biologically more realistic system, we performed AIMMD simulations of a mammalian plasma membrane (PM) mimetic[26,60]. We focused on the two most abundant lipid species: cholesterol and 1-palmitoyl-2-linoleoyl-sn-glycero-3-phosphocholine (PLPC). For the comparably apolar cholesterol, with a single hydroxyl group at its polar end, the AIMMD samplers quickly converge to a dry, pore-less tunnel mechanism (Fig. 5a and Supplementary Fig. 12). By contrast, the samplers for PLPC lipid with its zwitterionic phosphatidylcholine headgroup switch to a flip-flop mechanism closely mimicking that of the pure DSPC bilayer. In this pathway, the pores first destabilize so that PLPC translocates along narrow, transient water nanowires (Fig. 5b and Supplementary Fig. 13). Eventually, though, in most of the samplers, these nanopores collapse so that instead, only the lipid headgroup is solvated in a water nanodroplet, passing through the otherwise intact lipid bilayer.

## Discussion

With the recently developed AIMMD methodology[44], we conducted an extensive numerical study of unassisted flip-flop of lipids in fully atomistic representations of two model membranes and a plasma membrane mimetic, and for reference in a model membrane at coarse-grained resolution.

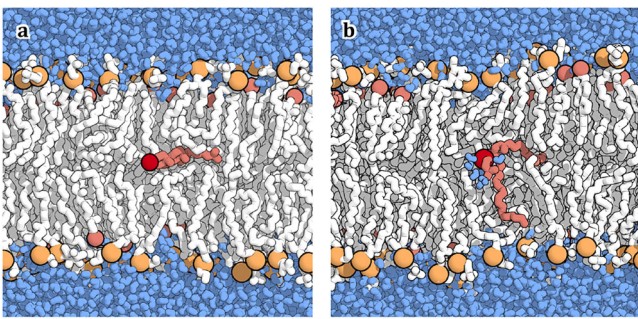

**Fig. 5 | Transition states states ($\phi \approx 0.5$) of lipid flip-flop in plasma membrane mimetic at atomistic resolution. a** Halfway across the membrane, cholesterol tends to lie flat at the membrane center. **b** The polar head of PLPC lipid tends to retain a small hydration shell. Lipids and water are shown as sticks, phosphorus and cholesterol oxygen as spheres.

For a bilayer of DMPC lipids in atomistic representation, flip-flop occurs predominantly by passage across spontaneously formed water nanopores. Once formed, the water nanopores typically stayed open long enough for multiple flip-flop events. As strong evidence for the dominance of the pore pathway $\Pi_P$ here, we first improved the statistics by running multiple TPS MC chains starting from different seed paths that jointly cover the two extreme mechanisms of a pre-existing pore and of dry lipid tunneling. Importantly, already the first paths in each chain were unbiased transition trajectories, albeit from a transition state (here, with a lipid at the bilayer center or with a pore) created by gently applying restraints. As new transition paths were discovered, memory of the seed paths was quickly lost (Supplementary Fig. 6c). We even observed that the character of the transition state changed: in all runs starting with dry tunneling $\Pi_T$, water pores formed eventually, leading to a $\Pi_P$ mechanism (Fig. 3a). This pathway via nanopores then persisted for all TPS walkers, ensuring the convergence to the unbiased, equilibrium TPE of our atomistic DMPC membrane.

AIMMD was also able to effectively sample water nanopore nucleation in an unbiased way. It confirmed that, first, the pore is established by a percolating water thread[48], which then allowed lipid headgroups to enter into the bilayer, with or without flip-flop. Pore formation thus appears to precede flip-flop, which occurs by chance in pores that live long enough, about 0.4 μs before pore collapse in our system (Supplementary Fig. 6).

We observed the other extreme case of pore-less flip-flop via the $\Pi_T$ pathway with the coarse grained DMPC Martini lipids and at the start of the all-atom TPS MC chains. Despite the involvement of an entire lipid patch in the flip-flop process, the committor network model was able to encode all relevant microscopic details. We found that to best predict the outcome of a ($\Pi_T$) transition, the model needed to take into account the surrounding network of lipids, most importantly their headgroups.

The dry lipid flip-flop mechanism observed for Martini DMPC lipids (Fig. 2) could be recapitulated for cholesterol in atomistic simulations of a plasma membrane mimetic (Supplementary Fig. 12). The committor for dry lipid passage is described well by a quasi-linear expression both for Martini and atomistic simulations (Supplementary Fig. 2 and Supplementary Fig. 8), with a linear projection of a large feature space entering a one-dimensional nonlinear function.

Most strikingly, we found that, after extensive training, our deep neural network with a ∼ 660-dimensional feature space encoded the committor in a nearly linear fashion. While neural networks are in general considered to be quasi black boxes able to approximate highly non-linear and hard-to-interpret functions, our network instead converged to a weighted average of distances to the neighboring lipids as an optimal reaction coordinate, associated with a simple uni-

directional reactive flux. This thought-provoking result connects to early linear models for $\phi$[47], as well as the idea of transition tubes[61] as approximately straight pathways through the transition region. The unanticipated tendency to linear models in sufficiently high dimensions is consistent with Cover's theorem[62] as a statement on the effectiveness of linear classifiers in high-dimensional spaces. By increasing the dimension of the feature space, linear models become more effective in discriminating configurations, here according to their committor values. However, the need to regularize the network representation of $\phi$ to prevent overfitting may play a role as well. How linear the transition funnel is close to the transition state emerges as an interesting future research direction well suited for the AIMMD method.

So, which transition mechanism of flip-flop is the correct one? Both atomistic and coarse-grained force fields are known to suffer from inaccuracies, which here may lead to the observed qualitatively different behavior of tunneling, with or without passenger water molecules, and pore mediated flip-flop. For DMPC lipids, the Martini case showed us how a lipid may flip without water (at least in part due to the well-known instability of Martini water pores[43,63]) but the all-atom representation instead leads to fully grown water pores to diffuse through. A middle-ground between a completely dry tunneling and nanopore formation might thus be what we observed for atomistic DSPC lipids and for PLPC lipids in the plasma membrane, where the rare local membrane thinning combined with narrow water threads and nanodroplets to establish a passageway for an even rarer lipid flip-flop.

From here having captured tunnel, pore, water-thread, and water-droplet mechanisms of flip-flop in closely related systems, we deduce that the relevant free energy barriers have comparable heights. The dominance of one or the other mechanism will then depend on system and condition, in line with earlier MD studies (see, e.g., refs. 24,31,37). For instance, lipids with large polar or highly charged headgroups may favor water nanopores even in a thick bilayer, where tunneling or water threads may dominate for a zwitterionic lipid. Also, a higher membrane bending rigidity (e.g., due to cholesterol) should suppress both pore formation and lipid flip-flop[64–66], as suggested here by the pronounced local bulging and thinning of the DSPC and plasma membrane, compared to the pore-forming DMPC bilayer. In biological membranes, scramblases relax bilayer asymmetries by providing comparably polar passageways for lipid headgroups for comparably fast lipid flip-flop[12,13,67]. Here, in neat membranes, functionally similar but highly transient polar passageways are provided by the fleeting appearance of water nanopores, nanowires, and nanodroplets. For cell membranes, we expect spontaneous phospholipid flip-flop unaided by proteins to occur via the mechanism we observed for neat DSPC bilayers and for PLPC lipids in the plasma membrane mimetic, i.e., with local membrane thinning and a transient water wire, without forming a metastable water nanopore. This mechanism can be considered intermediate between the extremes of dry lipid tunneling and wet water nanopore formation.

The connection between water pores and flip-flop is also coming into focus in experimental studies (see, e.g., refs. 16,68,69), having clear ramifications on the mechanistic interpretation of observations of lipid flip-flop-associated relaxation processes[19]. We expect that flip-flop mediated by water nanopores is essentially independent of headgroup size and charge of the flipping lipid. By contrast, flip-flop through dry tunnels should depend strongly on the size and charge of the headgroup, which partially loses its solvation shell during passage through the bilayer. By varying headgroups of the probe lipid and acyl-chain lengths of lipids in the background membrane, it should thus be possible to probe the transitions between different mechanisms of flip-flop, e.g., by estimating the entropy and enthalpy associated with defect density changing with temperature[16].

## Methods

### Molecular dynamics simulation

For MD simulations of the coarse-grained DMPC bilayer, we used gromacs version 2022[70] and the Martini 3[42,71] model (see Supplementary Fig. 14b, c for a sketch). A bilayer of $2 \times 225$ lipids was solvated in water with 0.15 mol/L NaCl in a $\sim 12 \times 12 \times 12$ nm$^3$ box using the insane.py[42] script. After energy minimization via gradient descent, the system was shortly equilibrated for 0.1 ns with 2 fs timestep, after which we performed a longer equilibration run with 20 fs time step for 1 μs, both in the semi-isotropic $NP_{xy}P_zT$ ensemble, using v-rescale thermostat[72] at 310.15 K with $\tau = 1$ ps (membrane and solvent coupled separately), and pressure couplings via Parrinello-Rahman[73] at 1 bar with $\tau = 12$ ps and $\kappa = 3 \times 10^{-4}$ bar$^{-1}$. Van der Waals interactions were handled with cutoff at 1.1 nm with potential-shift, Coulomb interactions via reaction field[74] with r = 1.1 nm with a dielectric constant of 15 and an infinite relative reaction-field dielectric. To test whether the reaction field electrostatics in our Martini simulations underestimated the headgroup desolvation penalty in the apolar center of the bilayer, we performed additional simulations with particle-mesh Ewald[75] (PME) electrostatics. Apart from setting the dielectric constant to 15, we left the TPS protocol unchanged. We found that the use of PME had no discernible effect on the observed flip-flop mechanism (Supplementary Fig. 15).

We also built a solvated all-atom DMPC bilayer (and similarly for DSPC lipids) using CHARMM-GUI[60,76], maintaining the initial a $12 \times 12 \times 12$ nm$^3$ box and using TIP3P water with 0.15 mol/L NaCl ions (Supplementary Fig. 14a). The double layer was modeled by the Charmm36 force field[77]. We also performed MD simulations of a mammalian plasma membrane mimetic. We downloaded the membrane model[26,60] from the CHARM-GUI archive and doubled the membrane area, resulting in a box of size $10.6 \times 10.6 \times 12$ nm$^3$. The resulting lipid numbers and mole fractions are listed in Supplementary Table 2. The MD simulations were performed with the same aqueous solvent composition, force field, equilibration sequence, and parameters as for the other systems.

The CHARMM-GUI schedule was set to a gradient descent minimization with position restraints of the lipids ($k = 1000$ kJ mol$^{-1}$nm$^{-2}$) and their joint dihedral ($k = 1000$ kJ mol$^{-1}$rad$^{-2}$), which was followed by an $NVT$ equilibration with the same restraints for 125 ps with 1 fs time step, with Berendsen thermostat[78] at $T = 310.15$ K (340.15 K in case of DSPC) with $\tau = 1.0$ ps (membrane and solvent coupled separately) and constrained hydrogen bonds (LINCS[79]). Van der Waals interactions were handled with cutoff at 1.2 nm, with force-switching from 1 nm, Coulomb interactions via PME[75] with $r = 1.2$ nm. Then followed 125 ps with $k = 400$ kJ mol$^{-1}$nm$^{-2}$ and 400 kJ mol$^{-1}$rad$^{-2}$, respectively, after that a 125 ps $NP_{xy}P_zT$ run at 1 bar with $\tau = 5$ ps and $\kappa = 4.5 \times 10^{-5}$ bar$^{-1}$, with $k = 400$ kJ mol$^{-1}$nm$^{-2}$ and 200 kJ mol$^{-1}$rad$^{-2}$, then 125 ps with 2 fs time step and $k = 200$ kJ mol$^{-1}$nm$^{-2}$ and 200 kJ mol$^{-1}$rad$^{-2}$, then 125 ps with $k = 40$ kJ mol$^{-1}$nm$^{-2}$ and 100 kJ mol$^{-1}$rad$^{-2}$, and then 125 ps without restraints. We then performed a 100 ns long simulation with 2 fs time step, v-rescale temperature coupling and Parrinello-Rahman pressure coupling.

### Transition path sampling

To test whether lipids prefer a spontaneous tunneling through the bilayer ($\Pi_T$), or the diffusion through formed water pores ($\Pi_P$), we set up initial transition pathways for these two cases (Fig. 1). Pathway $\Pi_P$ required the preparation of a water pore by introduction of a flat-bottomed position restraint on the lipids in the center of the simulation box (for the PM, we shift the center to have 8 different staring pores). To this end, we performed 1 ns (10 ns in case of Martini and the PM) of simulations with $k = 500$ kJ mol$^{-1}$nm$^{-2}$ and distance to the center $r$ ranging from 0.5 (head) to 1.6 nm (tail) to open the pore (for details see the Zenodo repository ref. 80). With fixed pore, we performed 10

ns (100 ns in Martini MD) of simulations, in which we also fixed one of the lipids chosen as probe lipid in the middle of the bilayer using an additional cylindrical harmonic restraint of the PO4 group (ROH of the PM cholesterol) with $r = 2$ nm, $k = 1000$ kJ mol$^{-1}$nm$^{-2}$. We used the last 1 ns (10 ns for Martini) as an initial trajectory to pool SPs for parallel TPS using AIMMD (see below). Using 8 samplers (6 for Martini), we ran a total of 100 MC TPS steps, i.e., 100 TPS simulations with fixed pore but unbiased probe lipid, of which we used for each sampler the last accepted one as seed for the following unbiased TPS. See Supplementary Fig. 14d for snapshots of one of these initial $\Pi_P$ paths. Preparation of initial $\Pi_T$ trajectories was achieved by a harmonic constraint pulling the probe lipid headgroup with $v = 0.001$ nm/ps, $k = 1000$ kJ mol$^{-1}$nm$^{-2}$, by simultaneously preventing water to enter the double-layer by use of a flat-bottomed position restraint of $k = 500$ kJ mol$^{-1}$nm$^{-2}$ and $r = 1$ nm from the mid-plane, resulting in a pore-free transition (Fig. 1, $\Pi_T$). We repeated this procedure to pull both upwards and downwards to have 4 + 4 (3 + 3 for Martini) seed paths (for the PM, we use a different lipid each time.). These rough transition pathways were then used for sequential TPS shooting. We ran a total of $N = 1000$ MC steps with water restraint and unbiased probe lipid. We used the last accepted one as seed for the following unbiased TPS. See Supplementary Fig. 14e for snapshots of one of these initial $\Pi_T$ paths. In both cases of initial starting transition pathways, we then performed unbiased (i.e., without flat-bottomed restraints) simulations to sample the transition state ensemble. We performed sequential two-way shooting TPS simulations via the AIMMD framework[44].

AIMMD aims for a high success rate of sampling flip-flop transitions by simultaneously estimating the corresponding committor $\phi(\mathbf{x}|\mathbf{w})$ via a neural network with weights $\mathbf{w}$. We predict from a set of microscopic input features $\mathbf{x}$ in what state the trajectory will end and from what state it came by minimizing the negative log-likelihood of shooting outcomes,

$$L(\mathbf{w}) = \sum_{i=1}^{N} \ln \left[ \binom{n}{k_i} \phi(\mathbf{x}_i^{\mathrm{SP}}|\mathbf{w})^{k_i} (1 - \phi(\mathbf{x}_i^{\mathrm{SP}}|\mathbf{w}))^{n-k_i} \right],$$

in terms of the weights $\mathbf{w}$ of the network, using as training the data of the so-far sampled $N$ MC steps in terms of their SP features $\mathbf{x}^{\mathrm{SP}}$ and number of times $k$ the propagated trajectory hits the final (e.g., state $\mathcal{U}$) state (where for two-way shooting we have $n = 2$ and $k_i \in \{0, 1, 2\}$). To accelerate the learning of the committor, we include the SPs of the initial restraint runs in the training set, see Supplementary Fig. 16 for convergence of the loss for the Marini case. We produce $N = 1000$ MC steps for $\Pi_P$ (12000 MC steps in case of Martini) and 2500 for $\Pi_T$ (10000 Martini). The estimate of $\phi$ is then used in the sequential TPS to efficiently sample SPs from the previous transition path. We allow for some deviations of shooting from the optimal $\phi = 0.5$ iso-surface by sampling from a Cauchy distribution of the logit $q$ of $\phi$ ($q = \ln\left[\frac{\phi}{1-\phi}\right] \sim \text{Cauchy}(\mu = 0, \gamma = 1)$). To this end, we estimate the actual distribution $P(q|\text{TP})$ of the TPS data (by a histogram of $q$), to reweigh each frame to a Cauchy sample. We choose SPs uniformly first, create the histogram after 100 MC steps, and update every 250 steps.

TPS of pore nucleation was seeded by extracting TPs from the $\Pi_T$ samplers transitioning to the $\Pi_P$ mechanism. We first sampled 1000 snapshots uniformly from all trajectory frames with pore reaction coordinate $0.5 < \xi_P < 1.0$. We then used AIMMD to uniformly pick one of these frames until a first trajectory was accepted. After that, we again performed sequential shooting, using 8 samplers with a total of 1000 MC steps, with the same shooting point selection criterion as before.

### Input features and network architecture

Using MDAnalysis[81], we define the two final states of the transition by the transversal displacement $z$ from the midplane (defined by the lipid

P atoms (PO4 bead for Martini) with respect to the vertical center ($z = 0$) of all P's). Based on the distribution of heads in the initial equilibrium simulations, we set the state $\mathscr{L}$ when $z < -1.3$ nm, and state $\mathscr{U}$ when $z > 1.3$ nm (1.7 nm for DSPC, 1.65 nm for cholesterol and 1.9 nm and PLPC of the plasma membrane), respectively.

To monitor the internal conformations, in addition to $z$, we also tracked the probe lipids radius of gyration and its tilt angle $\theta$ defined by the average distance vector to the P (PO4) atom and the $z$-axis.

For interactions with the other lipids, we also recorded the indentation of the upper and lower leaflet by the standard deviations from their respective centers.

We then tracked the displacement of each P (PO4) bead with respect to the probe sorted by distance, tracking the first 20 neighbors ($3 \times 20$ coordinates) to reduce noise. We also included the number of water molecules in the first and second shell around the probe P (PO4 in Martini), using the indicator function of ref. 82. As for the total number of water molecules inside the bilayer, we counted the number of water oxygens (W beads) within $\Delta z = 0.5$ of the mid-plane. See a detailed list of input features in Supplementary Table 3.

During the AIMMD runs, the neural network estimates the committor to the $\mathscr{U}$ state via a latent space representation $\phi(\mathbf{x})$, where we first selected 68 input features $\mathbf{x}$ to be encoded through 5 hidden layers. More specifically, the data is processed via a linear compression (with a small dropout probability during training), after which followed a ResNet[83] unit (with ELU activation) of depth 4. We do this to sequentially go from $68 \rightarrow 46 \rightarrow 31 \rightarrow 21 \rightarrow 14 \rightarrow 10 \rightarrow 1$, where at the last step we only use a linear unit. The output $q(\mathbf{x})$ is then transformed by a softmax to the probability $\phi$. See Supplementary Fig. 17a for a sketch and Supplementary Table 1. Note that the networks with $\sim 660$ features discussed in Results were used later in postprocessing, as described below.

The pore nucleation transitions were defined by the pore reaction coordinate $\xi_P$ from ref. 36, which combines the process of pore nucleation with that of pore expansion. The former is evaluated in terms of what fraction of the membrane (in terms of slabs along $z$ at the nucleus) is already occupied by polar atoms, the "pore-chain" $\xi_{ch}$[39]. The latter counts the number of water molecules inside a formed (assumed cylindrical) pore to estimate its radius $R$, and is added to $\xi_{ch}^s$ when close to 1, in units of the radius $R_0$ of a just fully nucleated pore. Here, we set the state boundary of the flat membrane, $\mathscr{F}$, to $\xi_P < 0.05$, and that of an expanded pore, $\mathscr{P}$, to $\xi_P > 2.0$. To study which features best describe the committor, we chose as input features of its neural network model all of $\xi_P$ and its constituents $\xi_{ch}$ and $R$.

The reported parameters for DMPC lipids induced an artificial meta-stable state in the transition region we accounted for; see Supplementary Fig. 18 and its caption. In case of Martini, we also changed the parameters of $\xi_P$ by decreasing the number of subdivisions to 4, with a cylinder size of $Z_{mem} = 1.8$ nm, $R_{cyl} = 1.0$ nm, counting the polar atoms for calculating the pore radius within $D = 1.2$ nm, as well as changing the switch towards pore expansion at $\xi_{ch}^s = 0.9$ with a radius $R_0 = 0.38$ nm. In this way, we aim to balance the noise around $\xi_P \approx 0$ with being able to detect water threads, as well as a smooth transition for large $\xi_P$.

We also feed in coordinates suggested by Bubnis and Grubmüller[40], who consider the distances of different atom types to the pore center. In our case, we use the pore center definition of ref. 39, a weighted circular mean of the headgroups. For the isotropic, lateral and axial distance to the center we measured the 1st, 2nd, and 3rd NN, as well as an average over the first 2, 3, 4, 5 and 10. For the axial distance, as detailed in ref. 40, we took the maximal average over neighboring pairs. We chose the same atom types, water O, P, N + P, N + P + $O_{H2O}$, carbon tails, and all carbons. In total, we end up with a network shape $147 \rightarrow 85 \rightarrow 50 \rightarrow 29 \rightarrow 14 \rightarrow 17 \rightarrow 1$. See also Supplementary Table 4.

## Accuracy of committor models

After the TPS production run, we tested if network architectures other than the initial one used in AIMMD resulted in a better committor estimate. To this end, we define the accuracy $\alpha$ of the committor model by the excess variance of committor estimates not explained by a binomial distribution. The probability $p_{bin}$ of $k$ hits of the final state with $n$ shots from a starting configuration with exact committor $P$ is

$$p_{bin}(k|n, P) = \binom{n}{k} P^k (1-P)^{n-k}.$$

We assume that $P$ is beta distributed around our estimate $\phi$ of the committor,

$$p_{beta}(P|a, b) = \frac{1}{B(a,b)} P^{a-1}(1-P)^{b-1},$$

which defines the Bayesian conjugate prior normalized by the beta function $B(a, b)$. We enforce the means to match, $\langle P \rangle = \phi$, by setting

$$a = \frac{\alpha}{1-\alpha}\phi, \quad b = \frac{\alpha}{1-\alpha}(1-\phi),$$

with a constant $\alpha$ in the range $0 \leq \alpha \leq 1$. The variance of $P$ in the beta distribution is then

$$Var[P] = (1-\alpha)\phi(1-\phi),$$

with its maximum and minimum at $\alpha = 0$ and 1, respectively. Convolving the binomial and beta distributions gives the probability to see $k$ hits,

$$p(k|n, \phi, \alpha) = \int_0^1 dP\, p_{bin}(k|n, P) p_{beta}(P|a, b) = \binom{n}{k}\frac{B(w\phi + k, w(1-\phi) + n - k)}{B(w\phi, w(1-\phi))},$$

which is a beta-binomial distribution of $k$, where $w = \alpha/(1-\alpha)$

We now treat $\tilde{p}(\alpha|k, n, \phi) \propto p(k|n, \phi, \alpha)$ as a Bayes posterior for the accuracy $\alpha$, having treated $P$ as a nuisance parameter.

Given a sample of shooting data, $(\phi_i, k_i, n_i)_{i=1}^N$, where $\phi_i$ is the committor predicted by the model, we accordingly estimate the accuracy $\alpha$ of the committor model by maximizing the log-posterior,

$$L(\alpha) = \sum_{i=1}^N \ln\left[p(k_i|n_i, \phi_i, \alpha)\right],$$

with respect to $\alpha$. For $\alpha = 1$, the committor model fully explains the data, $\phi_i = P_i$ for all $i$; for $\alpha = 0$, the data are best explained by a combination of fully committed states, $P = 0$ and $P = 1$, indicating a complete lack of predictive power.

We make a bootstrap estimate of $\alpha$ by repeating 10 times: split the data into training (all but one MC chain) and validation set (that chain), train the model for some number of epochs, and then pick 100 times bootstrap samples (with replacement) from the validation set to estimate the validation loss and accuracy. We used the validation loss to set the number of epochs to prevent over-fitting. We tested the influence of the number of hidden layers, number of nodes, as well as dropout. See Supplementary Table 1 and Supplementary Fig. 17b for all tested networks.

As a final test of systematic error of the network's predictions $\phi_i$, we performed additional committor shots now with $n_i = 20$ (instead of the $n_i = 2$ during TPS) for an estimate of the actual committor, $P \approx k_i/n_i$ (Supplementary Fig. 7a, b). To assess bias and variance, we try to avoid the binning used previously[44], and instead fit a line to the actual

logit ($\ln k/(n-k)$) vs. the logit predicted by the model ($\ln \phi/(1-\phi)$),

$$q^{\mathrm{lin}} = \ln \frac{\phi^{\mathrm{lin}}}{1-\phi^{\mathrm{lin}}} = c \ln \frac{\phi}{1-\phi} + d, \; c_{\mathrm{fit}}, d_{\mathrm{fit}} = \underset{c,d}{\arg\min} \sum_{i=1}^{N} \left( q_i^{\mathrm{lin}} - \ln \frac{k_i}{n_i - k_i} \right)^2.$$

Together with the estimate for $\alpha$, we can visualize the spread of the data by the standard error of the mean, $\Delta \phi^{\mathrm{lin}} = \sqrt{1 - \alpha\left(1 - \frac{1}{n}\right)} \sqrt{\phi^{\mathrm{lin}}\left(1 - \phi^{\mathrm{lin}}\right)}$.

We also tested additional input features of the network not used in the initial network models (which had used only 68 features). While in the initial model we simply averaged all P (PO4) atoms to define the midplane, we got better predictions by using the lipid tail atoms weighted (using the sigmoid of ref. 84) by their distance to the probe in the *xy*-plane, see Supplementary Fig. 17b. We also refined the definition of the tilt angle $\theta$ by calculating the normal to the membrane via the P (PO4) atoms of the two leaflets, weighted by distance to the probe.

We also track the neighboring lipids for upper and lower leaflet separately and use the 10 nearest neighbors, respectively. For the trajectories, to prevent the switching of rankings of the lipids, we calculate the time-averaged distance to the probe to define the identity and rank of the 10 most important lipids on the upper and lower leaflet, respectively. Lastly, we also track the $10 \times 10$ distance matrix between 10 central atoms equivalent to the 10 beads of Martini (and the 10 beads in case of Martini) of the $3+3$ closest lipids and the beads of the probe.

The internal state of the lipid probe seemed of minor importance. In Martini, we only considered the distances $d(C_{3A}, C_{3B})$ and $d(C_{2A}, C_{2B})$, as well as the angles $\angle C_{3A} GL_1 C_{3B}$, $\angle P_{O4} GL_1 C_{1A}$ and $\angle P_{O4} GL_2 C_{1B}$, which together are able to describe a split of the two lipid tails. A detailed description of these features can be found in Supplementary Table 5. We proceeded to use these improved features, which are the ones discussed in Results.

While training a neural network model with a total of 663 (667 Martini) input features, we added L2 regularization to prevent overfitting, allowing for a larger number of training epochs. Since this may inevitably suppress strong nonlinearities in the model, we then took the learned reactive flux vector $\mathbf{v} = \nabla \phi / |\nabla \phi|$, projected the data onto $\mathbf{x} \cdot \mathbf{v}$, and then again trained a 3-dimensional committor model using only $z, \xi_{\mathrm{P}}$ for atomistic MD and $\theta$ for Martini MD, and $\mathbf{x} \cdot \mathbf{v}$ as inputs to test the consistency of the linear model. In Results, we discuss and contrast the resulting high and low-dimensional committor models.

## Water pore lifetime

Starting from shooting point structures of the last MC step of the DMPC pore-mediated flip-flop, we started free, unbiased simulation runs, using the same parameters as before, for either a maximum of 200 ns or until collapse of the pore was observed. We then calculated the mean pore lifetime using a maximum-likelihood estimate for randomly censored data with exponential kinetics, $\tau = \sum_{i=1}^{N} t_i/n$, where $t_i$ is the duration the pore stayed open in simulation run $i$, either before closing spontaneously in $n$ of the $N$ runs or before the run was terminated with still intact pore in the remaining $N - n$ runs. One recognizes in the estimator the ratio of the aggregate time of being uninterrupted in the open state divided by the number of closing transitions. This estimator maximizes the likelihood $L(k) = \prod_{i=1}^{n} p(t_i|k) \prod_{i=n+1}^{N} S(t_i|k)$ written as a product of the survival probability $S(t|k) = e^{-kt}$ (for terminated runs) and the corresponding probability density $p(t|k) = -dS/dt = k e^{-kt}$ (for runs in which the pore closed).

## Estimation of projected densities

All TPE densities were estimated by projecting the MC chain data to the respective observables and evaluating the histogram. In case of the Martini DMPC flip-flop, those data did not include the initial relaxation of $\Pi_{\mathrm{P}}$ samplers to the $\Pi_{\mathrm{T}}$ mechanism. Those data are shown in Supplementary Fig. 1b, using a $k$-neigherst neighbor estimate: for each

point in $z, \xi_{\mathrm{P}}$, we measure the radius $r$ of the smallest circle including $k$ data and estimate the density as $\sim r^{-2}$.

To visualize the committor isosurfaces in 2D projections onto coordinates $s, t \in \{z, z_1, \xi_{\mathrm{P}}, \Delta \mathbf{r}^{\mathrm{all}} \cdot \mathbf{v_r}\}$, we estimated the local average input features to the committor network, $\langle \mathbf{x} \rangle_{s,t}$, from the $k$-nearest neighbors. We then projected the average onto $\mathbf{v}$ and evaluated the quasi-linear committor model, $\varphi(\langle \mathbf{x} \rangle_{s,t} \cdot \mathbf{v})$.

## Reporting summary

Further information on research design is available in the Nature Portfolio Reporting Summary linked to this article.

## Data availability

Source data including features and network models are made available in the Zenodo repository [https://doi.org/10.5281/zenodo.14836411] ref. 80.

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

## Acknowledgements

This work was supported by the Max Planck Society. We thank Balázs Fábián, Hendrik Jung, Stefan Schäfer and Jürgen Köfinger for discussion and comments on the manuscript, Vida Lotufo for assistance with ana-lysis of DSPC lipids and the Max Planck Computing and Data Facility for computational resources.

## Author contributions

G.H. conceived the project, M.P. performed the MD simulations, M.P. analyzed the data, M.P. and G.H. wrote the manuscript, G.H. supervised the project.

## Funding

## Competing interests

The authors declare no competing interests.
