## [Transparent Peer Review file · Nature Communications]

AI-guided transition path sampling of lipid flip-flop and membrane nanoporation

Corresponding Author: Professor Gerhard Hummer

Version 0:

Reviewer comments:

Reviewer #1

(Remarks to the Author)

Post and Hummer have written a paper that computationally elucidates the mechanism of a process occurring in cell membranes, the significance of which for the function of cell membranes is indisputable. Spontaneous translocation of lipids (flip-flop) is, on the one hand, a harmful process, because its correction consumes the energy reserves of the cells. On the other hand, translocation events play a central role in certain cellular functions, such as programmed cell death, although it is not entirely clear whether these events are controlled by specific membrane proteins. Experimentally, detailed study of translocation events is somewhat impossible, because they are exceptionally rare events, and because the time required for their occurrence is typically on the order of 10-100 ns. On the other hand, computational research is also challenging for the same reasons.

In this paper, the problem and its analysis are approached from a new perspective. Instead of spontaneous events, the focus of the research is on the transition state halfway through the translocation event, and by studying this state, using new methodology based on artificial intelligence (AI) tools, neural network-based models have been developed and used to predict the most obvious candidate for the lipid translocation mechanism.

Previous research has found that lipid translocation occurs, among other things, in connection with the formation of membrane pore defects. Post and Hummer report in their paper that this is the most obvious realistic candidate for lipid translocation. The result is not very surprising. However, what makes the paper important is the AI-based methodology presented in the manuscript, which could be used to investigate similar phenomena in detail in follow-up projects.

The methodological expertise described in the paper is undeniably excellent and brings a significant step forward in the field.

What is less fascinating about the paper is the chosen research topic. DMPC is not biologically very important, so it is unclear why it was chosen as a research target.

Furthermore, the paper gives considerable space to the results of the coarse-grained Martini model, which predicts a mechanism which is inconsistent with the results of atomic-level simulations (and, in the reviewer's view, quite likely also with experimental results). The Martini model sometimes works quite well, but it is well known that its results are also misleading in many processes. In the case studied in the paper, it is not clear whether the description of electrostatic interactions (reaction field) used in the Martini model is realistic enough to describe the bottleneck of the translocation event, i.e. the pulling of the charged/polar lipid head group into the lipid membrane. The authors should seriously consider whether the article would be better (and less contradictory) if the Martini part were removed from the paper in its entirety. (It can be published as a separate paper if necessary, highlighting e.g. the development needs of the Martini model.)

The Martini model proposes that lipid translocation would occur without significant membrane reorganization (pore defect formation). For PC-based lipids, where the head group dipole is significantly large, atomic-level simulations do not predict this. The exception is sterols, where the head group (often the OH group) dipole moment is weak enough that the sterol can embed itself inside the membrane without pore formation. Another and more biologically intriguing context is the translocation of lipids (including PC lipids) along the surface of scramblase proteins. For GPCR proteins (opsins, among others) it has been observed that lipids are able to translocate along their surface without pore defect formation. The PC

head group then pulls some water string into the membrane but quickly replaces it with interactions between the PC head group and amino acids on the protein surface, thus allowing translocation to proceed without a large free energy barrier. If the authors wish to present and clarify a pore-independent translocation mechanism in their paper, these would be potential contexts.

In summary, the paper would benefit from a clearer statement of which part of the presented research is exceptionally novel and significant. The reviewer feels that the methodology section presented in the paper answers this question positively; the biological context less so.

Reviewer #2

(Remarks to the Author)

The manuscript by Matthias Post and Gerhard Hummer provides valuable insights into the mechanisms of lipid flip-flop and nanopore formation. The authors apply transition path sampling (TPS) within the Artificial Intelligence for Molecular Mechanisms (AIMMD) framework, recently introduced by the same research group. By comparing coarse-grained force fields (Martini 2) with all-atom simulations (CHARMM36), lipid tail length, and the role of the pore formation process, the study offers a detailed perspective on the mechanism of lipid flip-flop without the use of biasing potentials, which were commonly employed in previous studies relying on enhanced sampling methods.

The topic is highly relevant and grounded in modern computational methodologies. The results are of broad interest to the community. However, the manuscript, in its current form, lacks clarity in several areas. Additionally, certain methodological aspects require further discussion and justification. Given the significance and quality of the study, I recommend publication after revisions addressing the issues outlined below.

Comments:

1. Page 2: The authors state: "As coordinate for the presence and size of a possibly associated pore, we use ξ_P by Hub [36]." Is the rationale for selecting this collective variable (CV) discussed? The work by Bubnis and Grübmueller (10.1016/j.bpj.2020.10.037) is mentioned but not elaborated upon, while the work by Rivel and coworkers was omitted (10.1021/acs.jcim.4c01960)—should the authors clarify why Hub's collective variable (CV) was chosen over others? Particularly, the alternative collective variables seem to point the influence of the lipid acyl chains in the pore formation process. How could this relate to the defect formation process assisting lipid flip-flop?
2. Page 4: Before referring to Figure 2b, it would be appropriate to introduce explicitly the different features, as well as the related models, potentially in a visual way. In the current version, the manuscript lacks clarity on these definitions, as not all the features are introduced in the text. Consequently, the reader should refer to Figure 2 caption, which itself refers to Table 1 (which is not mentioned in the main text), while looking to the details given in the Methods section where the reader is referred to Table S1. This brings unnecessary complexity. As an example, the sentence "We find that a full description of the tunnel transition mechanism requires the relative distances, $x = \Delta_{\text{rall}}$ (red), of beads of the lipid neighbor network." is the first reference to Δ_{rall} . The referenced figure does not explicitly mention $x = \Delta_{\text{rall}}$, and the color of the Nall model appears to be orange instead of red. The confusing definitions of the features and models render the manuscript rather hard to read. After providing a clear introduction of the features and related models, it would be advisable to check for the consistency in the terminology throughout the manuscript to ensure clarity. It may be necessary to leave in the Methods only the formal description of the input features (and to refer explicitly to the given subsection in the Results section, to guide the reader).
3. Page 4: The authors underlined that: "This implies a simple shape of the committor, $\Phi \approx \phi(x \cdot v)$, i.e., a linear model of the input features.". Could the authors elaborate on this assumption in relation to Figure 2, particularly the isolines in subplot e and/or subplots c and d, as this may reinforce their statement?
4. Page 4: The authors wrote: "Yet, most of the weight is found in the distances of the probe head to neighboring lipids (Fig. S2)". It is unclear how this is inferred from Figure S2.
5. Page 5: The authors estimate nanopore persistence time as $\sim 0.4 \mu\text{s}$ based on Figure S6a. The methodology relies on counting closure events in 200 ns trajectories. Could the authors discuss whether this method provides a maximum likelihood estimate? Additionally, how does this compare with estimates and methodology in Bennett et al. (<https://doi.org/10.1016/j.bpj.2013.11.4486>, e.g. Fig. 5)? The $0.4 \mu\text{s}$ estimate is also cited on pages 7 and 8—these references may need revision.
6. Page 6: The authors attribute the lower accuracy of the atomistic model compared to the coarse-grained model to sample size. Does this also explain the larger dispersion in the accuracy reported on Figure 3 (compared to that of Figure 2)?
7. Page 9: The statement "The middle-ground might thus be what we observed for atomistic DSPC lipids, where local membrane thinning combined with narrow water threads to establish a passageway for lipid flip-flop.". However, earlier results suggest that in DMPC, flip-flops are rare during pore nucleation and do not occur in the presence of narrow water threads. Does this truly represent a middle ground?
8. Page 10: "Pathway PIP required the preparation of a water pore by introduction of a flat-bottomed position restraint of the lipids in the center of the simulation box.". The authors chose to use Hub's CV as a measure of the pore state but decided to use inverted flat-bottom potentials to create the membrane pore, which is unlikely to follow a minimal free energy path (MFEP) during the pore formation process in the membrane. Could the authors comment on this choice?
9. Page 10: "To this end, we performed 1 ns (10 ns in case of Martini) of simulations with $k = 500 \text{ kJ mol}^{-1} \text{ nm}^{-2}$ and r ranging from 0.5 (head) to 1.6 nm (tail) to open the pore.". There is a lack of details to reproduce this procedure. Particularly, it seems that the change in the radius of the flat portion of the potential is not increased continuously; however, it is not mentioned how many steps were necessary to reach a 1.6 nm radius. The authors do not show that the system had time to relax after increasing the pore radius, neither between two steps used to increase the pore radius, nor after the last one (which is 10 times longer than intermediate steps) that is used to produce the shooting points for the TPS procedure. Using CV designed to follow the MFEP of pore formation, both Hub et al. and Rivel et al. point out that at least longer simulation times are necessary ($> 100 \text{ ns}$) for AA simulations at a given pore size to obtain a satisfying sampling for an umbrella

sampling procedure.

10. Page 10: "We used the last 1 ns (10 ns for Martini) to pool shooting points for parallel TPS using AIMMD (see below)". Were the shooting points selected randomly? At an even step? Is that enough to avoid any memory effect between frames?

11. Page 10: The authors mention that "[p]reparation of initial Π T trajectories was achieved by a harmonic constraint pulling the probe lipid headgroup with $v = 0.001$ nm/ps, $k = 1000$ kJ mol⁻¹ nm⁻² [...]", while for pathway Π P, the authors mention that they "performed 10 ns (100 ns in Martini MD) of simulations, in which [they] also fixed the probe lipid in the middle of the bilayer using an additional cylindrical harmonic restraint of the PO₄ group with $r = 2$ nm, $k = 1000$ kJ mol⁻¹ nm⁻²". Two different strategies seem to have been used to bring the probe lipid close to the tipping point. A steered MD procedure is detailed concerning the preparation of initial Π T trajectories, but the preparation of Π P is not as detailed. Could the authors add a description of the latter and comment on the difference between the procedures?

12. Page 11: The authors state that they used "MDAnalysis to define the two final states of the transition by the transversal displacement z from the midplane (defined by the lipid P atoms (PO₄ bead for Martini) w.r.t. the vertical center ($z = 0$) of all P's), namely the state L when $z < -1.3$ nm, and state U when $z > 1.3$ nm, respectively.". However, they don't comment on the motivation for the choice of such an arbitrary threshold. Particularly, how using a unique threshold regardless of the studied lipid type (DMPC v. DSPC) can influence the results?

Minor comments / typo:

1. Page 3: (minor/typo) The phrase "referring to Methods for details of the TPS setup" is vague given the number of subsections in the Methods. It would improve readability to guide the reader more specifically.

2. Page 4: (minor/typo) The sentence "To further explain how they instead tunnel through the bilayer, we study the importance of individual microscopic features describing the committor $\Phi(x)$." may read more fluidly if rephrased e.g.: "[...] of individual microscopic features x in describing the committor $\Phi(x)$."

3. Page 4: (minor/typo) The term "transition times" is introduced in "Notably, in this intermediate, unstable period, the corresponding transition times are the smallest, even compared to Π T (see Fig. S1c in the SI)." However, at this point in the manuscript, it is unclear how transition times are defined. It would be recommended to define this term, or to refer where we could find its definition.

4. Pages 3-4: (minor/typo) The manuscript states, "The flip-flop transition of an individual lipid (the 'probe lipid') is tracked by its transversal displacement z from residing in the lower leaflet (state L) to the upper leaflet (state U). Here, we time-order the TPs as $L \rightarrow U$ ". However, Figure S1a referred in the following paragraph illustrates a flopping event ($U \rightarrow L$). I am not sure whether it is necessary to indicate that the TPs are time-ordered, as I did not see where this information was relevant later in the manuscript. On the other hand, this may bring confusion to the reader.

5. Page 4: (minor/typo) The manuscript states, "After about $n \approx 500$ MC steps, we have a behavior similar to Π T, and thus, all water beads are flushed out and the pore is closed completely during the remaining transitions.". However, in Figure 2a, the closure appears to occur closer to 600 or 700 MC steps. Please verify.

6. Page 4: (minor/typo) The authors refer to " $\Delta r_{\text{all}} \cdot v_r$ " as the "mean neighbor distance," while it seems to be the projection of this quantity on v_r .

7. Page 4: (minor/typo) "E.g. a conformation with joined tails [...]" . It may be advisable to avoid beginning the sentence with "E.g."

8. Page 6: (minor/typo) "If we do, see Figure 3c, we see that again the committor prediction using distances Δr_{all} to all neighboring lipid atoms (yellow) outperforms a simple model using z alone (light blue)." Reconsider the phrasing; some commas may be missing.

9. Page 9: (minor/typo) Indicate the unit when discussing salt concentrations.

10. Page 9: (minor/typo) "after which we performed a" → add a space.

11. Page 9: (minor/typo) The statistical ensemble used in the short equilibration run is not clearly defined. Please provide additional details.

12. Page 10: (minor/typo) "and r ranging from 0.5 (head) to 1.6 nm (tail) to" → define explicitly what " r " represents.

13. Page 10: (minor/typo) "See Fig. Fig. S12e for snapshots of one of these initial" → Typo.

14. Page 10: (minor/typo) The authors provide the equation of the loss function used in the training of the neural network involved in the AIMMD framework. It seems that the sum from $i=1$ to N refers to the N features. That is not defined explicitly. Meanwhile, in the same paragraph, the authors indicate that they produce " $N = 1000$ MC steps for Π P".

15. Page 10: (minor/typo) "To accelerate the learning of the committor, we include the SPs of the initial restraint runs to the training set." The abbreviation "SP" for shooting point was not defined earlier.

16. Page 11: (minor/typo) "w.r.t." → It may be recommended to not use such an abbreviation.

17. SI (minor/typo) In the legend of Fig. S1 (and some others) in the SI, the authors mention that the subfigure "a" represents an "exemplary trajectory." The authors may consider rewording it or explaining in what manner this trajectory was selected to be particularly representative.

18. SI (minor/typo) In the legend of Fig. S1: "k-neighrest neighbor" → "k-nearest neighbor."

Reviewer #3

(Remarks to the Author)

The authors present a detailed mechanistic study of lipid flip flop transitions in both atomistic and coarse-grained forcefields for DMPC and DSPC lipid membranes. In particular, the use of an AI-guided transition path sampling approach provides detailed mechanistic description at the atomistic scale of the order of events along the lipid flip flop transition pathway. The application of transition pathway analysis of this system, particularly at the atomistic scale, would be of interest to the biomolecular simulation and rare events community. As pointed out by the authors, these results may hint at a more general linearization effect in high dimensional feature spaces, although this is not conclusively established in this work.

I have a few questions for the authors to potentially improve the manuscript.

1. How well are the neural network model weights fitted? I recommend adding a supplementary figure demonstrating convergence of the NN weights as measured by minimization of the loss function over training epochs, or some suitably related metric.
2. For the Charmm36 DMPC model described in Figure 3, what is the likelihood of a reverse transition of the TPS process back to the dry transition tube? It seems clear from this data that the dry pathway is a locally metastable transition tube which eventually transitions to the wet transition tube via the TPS process. However, the average number of MC steps required to make this transition from the dry to the wet tube seems on average much longer than the wet-initialized tubes MC processes were simulated. If these were extended, would the reverse process be observed and with what relative probability? Since the global stability of the wet transition process is a key result of this paper, I would suggest extending the TPS sampling or otherwise quantifying the stability of the wet process for the atomistic case.
3. Further, showing that one transition tube is meta-stable and the other is not meta-stable only qualitatively supports the conclusions, but I would suggest that the relative path probabilities of one tube over the other would more precisely quantify and support this point. A similar argument could be suggested for quantifying the relative probabilities of the order of molecular events sampled in the atomistic DMPC case. Since this evidence is central to the conclusions of the paper, it may be worth adding this analysis to strengthen the conclusion.

Minor suggestions to improve the presentation:

1. For me, it would help to enlarge or rearrange the text placed inside figure panels and on the axis markings in figures 2,3 and 4. I had a hard time reading them at first.
2. Although an extensive presentation of AIMMD is beyond the scope here, a short summary of the AIMMD method would be helpful.
3. I believe the methods details for the DPSC model is missing and should be added.

Version 1:

Reviewer comments:

Reviewer #1

(Remarks to the Author)

During revision, the manuscript has substantially improved. The methodology presented in the paper is an excellent addition to the toolkit and supports the acceptance of the manuscript.

There remains some degree of discrepancy between the atomic-level simulation results and the results of coarse-grained models (DMPC). In this respect, the revised paper clearly raises the issue and justifies the discussion of the coarse-grained model results by stating that it can be used to demonstrate an AI-based analysis of the "dry" mechanism. It is also true, just as the paper states, that there is no convincing experimental evidence that lipid flip-flop occurs in DMPC membranes through pores, or without them.

However, the authors should consider addressing one more theme in their paper. Experimentally, first of all, it is known that the means used by cells to protect their energy reserves, which are based on differences in salt ion concentrations across the plasma membrane, is cholesterol: when the membrane cholesterol concentration is increased, the formation of pores can be prevented (as confirmed by electroporation experiments). On the other hand, it is known that without cholesterol, cell membranes are very elastic (low bending rigidity), while at high cholesterol concentrations, cell membranes become more rigid (high bending rigidity). From this experimental observation, one could indirectly conclude, or at least make an educated guess, that in elastic cell membranes, lipid flip-flop is mainly based on pores, while in rigid cell membranes, lipid flip-flop does not require the presence of pores. This may also partly explain the MARTINI model results described in the paper, considering that the ability of the MARTINI model to describe the elasticity of membrane structures is not quantitatively very convincing.

Reviewer #2

(Remarks to the Author)

The authors have carefully addressed all my comments and remarks. Moreover, they extended the work to a realistic membrane model, which represents a substantial and valuable addition to the original manuscript.

The article was already of great importance and well-founded in its initial version. After the modifications, the manuscript is easier to follow, and the message is both clearer and stronger.

The authors should be commended for the quality of this work and the thoroughness of their revisions. I fully recommend the acceptance of this article in its current form.

Reviewer #3

(Remarks to the Author)

The authors present a detailed mechanistic study of lipid flip flop transitions in both atomistic and coarse-grained forcefields for DMPC lipid membrane, DSPC lipid membrane and a plasma membrane mimetic. In particular, the use of an AI-guided transition path sampling approach provides detailed mechanistic description at the atomistic scale of the order of events along the lipid flip flop transition pathway. The additions the authors have made in the revised version have significantly strengthened the manuscript I would recommend publication in it's current state.

RESPONSE TO REVIEWER COMMENTS

Response to Reviewer #1:

Post and Hummer have written a paper that computationally elucidates the mechanism of a process occurring in cell membranes, the significance of which for the function of cell membranes is indisputable. Spontaneous translocation of lipids (flip-flop) is, on the one hand, a harmful process, because its correction consumes the energy reserves of the cells. On the other hand, translocation events play a central role in certain cellular functions, such as programmed cell death, although it is not entirely clear whether these events are controlled by specific membrane proteins. Experimentally, detailed study of translocation events is somewhat impossible, because they are exceptionally rare events, and because the time required for their occurrence is typically on the order of 10-100 ns. On the other hand, computational research is also challenging for the same reasons.

In this paper, the problem and its analysis are approached from a new perspective. Instead of spontaneous events, the focus of the research is on the transition state halfway through the translocation event, and by studying this state, using new methodology based on artificial intelligence (AI) tools, neural network-based models have been developed and used to predict the most obvious candidate for the lipid translocation mechanism.

Previous research has found that lipid translocation occurs, among other things, in connection with the formation of membrane pore defects. Post and Hummer report in their paper that this is the most obvious realistic candidate for lipid translocation. The result is not very surprising. However, what makes the paper important is the AI-based methodology presented in the manuscript, which could be used to investigate similar phenomena in detail in follow-up projects.

The methodological expertise described in the paper is undeniably excellent and brings a significant step forward in the field.

RESPONSE: We thank the Reviewer for the very positive assessment and for the helpful comments and criticisms, as addressed below.

What is less fascinating about the paper is the chosen research topic. DMPC is not biologically very important, so it is unclear why it was chosen as a research target.

RESPONSE: We chose DMPC lipids as our main target for two reasons. First, lipid flip-flop in DMPC bilayers has been studied experimentally more than for other lipids. Second, and related to point one, the short acyl chains lead to comparably thin bilayers that facilitate lipid flip-flop. In the revised manuscript, we emphasize the model character of this system by reformulating in the introduction: “We apply this general framework first to neat DMPC lipid bilayers, as a single-species model used extensively in systematic studies of various membrane properties, including lipid flip-flop.”

To directly address the concern, we also performed additional simulations of an asymmetric plasma membrane mimetic (<https://pubs.acs.org/doi/10.1021/acs.jcim.1c01514>). We studied the flip-flop of the two most abundant lipid species: cholesterol and PLPC. In the Introduction, we alert the reader of these new findings:

“We confirmed that “dry” tunneling predominates for cholesterol flip-flop in MD simulations of a plasma membrane mimetic with leaflet asymmetry, whereas PC lipids cross between leaflets both along transient water nanowires and solvated by small water nanodroplets.”

In the Results section, we added the following paragraph:

“Dry and wet flip-flop in plasma membrane. To study lipid flip-flop in a biologically more realistic system, we performed AIMMD simulations of a mammalian plasma membrane (PM) mimetic. We focused on the two most abundant lipid species: cholesterol and PLPC. For the comparably apolar cholesterol, with a single hydroxyl group at its polar end, the AIMMD samplers quickly converge to a dry, pore-less tunnel mechanism (Fig. 5a and Supplementary Figure 12). By contrast, the samplers for PLPC lipid with its more zwitterionic phosphatidylcholine headgroup converge to a flip-flop mechanism closely mimicking that of the pure DSPC bilayer. In this pathway, the pores first destabilize so that PLPC

translocates along narrow, transient water nanowires (Fig. 5b and Supplementary Figure 13). Eventually, though, in most of the samplers, these nanopores collapse so that instead, only the lipid headgroup is solvated in a water nanodroplet, passing through the otherwise intact lipid bilayer.”

In the Methods section, we added:

“We also performed MD simulations of a mammalian plasma membrane mimetic. We downloaded the membrane model from the CHARM-GUI archive and doubled the membrane area, resulting in a box of size $10.6 \times 10.6 \times 12 \text{ nm}^3$. The resulting lipid numbers and mole fractions are listed in Supplementary Table 2. The MD simulations were performed with the same aqueous solvent composition, force field, equilibration sequence, and parameters as for the other systems.”

Furthermore, the paper gives considerable space to the results of the coarse-grained Martini model, which predicts a mechanism which is inconsistent with the results of atomic-level simulations (and, in the reviewer's view, quite likely also with experimental results). The Martini model sometimes works quite well, but it is well known that its results are also misleading in many processes. In the case studied in the paper, it is not clear whether the description of electrostatic interactions (reaction field) used in the Martini model is realistic enough to describe the bottleneck of the translocation event, i.e. the pulling of the charged/polar lipid head group into the lipid membrane. The authors should seriously consider whether the article would be better (and less contradictory) if the Martini part were removed from the paper in its entirety. (It can be published as a separate paper if necessary, highlighting e.g. the development needs of the Martini model.)

RESPONSE: We thank the Reviewer for these thoughtful comments and suggestion. After giving it much thought, we feel that including the coarse-grained results is important. First, they nicely represent an extreme form of a mechanism. We now show for cholesterol in the plasma membrane mimetic that this “dry” lipid passage is at play for lipids with weakly polar head groups, and we expect it to be relevant for thicker bilayers and at low water activity. Second, the Martini simulations led us to a quasi-linear form of the committor in a high dimensional feature space. We now discuss these points in more detail:

“The dry lipid flip-flop mechanism observed for Martini DMPC lipids (Fig. 2) could be recapitulated for cholesterol in atomistic simulations of a plasma membrane mimetic (Supplementary Figure 12). The committor for dry lipid passage is described well by a quasi-linear expression both for Martini and atomistic simulations (Supplementary Figure 2 and Supplementary Figure 8), with a linear projection of a large feature space entering a one-dimensional nonlinear function.”

To get a better understanding of the source of the discrepancy between atomistic and coarse-grained representations, we also ran coarse-grained MD simulations with particle-mesh Ewald electrostatics instead of reaction field electrostatics. We found virtually no difference to the previous results. We now state in the Methods section of the manuscript:

“To test whether the reaction field electrostatics in our Martini simulations underestimated the headgroup desolvation penalty in the apolar center of the bilayer, we performed additional simulations with particle-mesh Ewald (PME) electrostatics. Apart from setting the dielectric constant to 15, we left the TPS protocol unchanged. We found that the use of PME had no discernible effect on the observed flip-flop mechanism (Supplementary Figure 15).”

The Martini model proposes that lipid translocation would occur without significant membrane reorganization (pore defect formation). For PC-based lipids, where the head group dipole is significantly large, atomic-level simulations do not predict this. The exception is sterols, where the head group (often the OH group) dipole moment is weak enough that the sterol can embed itself inside the membrane without pore formation. Another and more biologically intriguing context is the translocation of lipids (including PC lipids) along the surface of scramblase proteins. For GPCR proteins (opsins, among others) it has been observed that lipids are able to translocate along their surface without pore defect formation. The PC head group then pulls some water string into the membrane but quickly replaces it with interactions between the PC head group and amino acids on the protein surface, thus allowing translocation to proceed without a large free energy barrier. If the authors wish to present and clarify a pore-independent translocation mechanism in their paper, these would be potential contexts.

RESPONSE: We thank the Reviewer for these insightful comments. In response, we expanded the discussion to connect our findings to scramblase function in biological membranes:

“In biological membranes, scramblases relax bilayer asymmetries by providing comparably polar passageways for lipid headgroups for comparably fast lipid flip-flop. Here, in neat membranes, functionally similar but highly transient polar passageway are provided by the fleeting appearance of water nanopores, nanowires, and nanodroplets.”

In summary, the paper would benefit from a clearer statement of which part of the presented research is exceptionally novel and significant. The reviewer feels that the methodology section presented in the paper answers this question positively; the biological context less so.

RESPONSE: We followed the suggestions by emphasizing the biological relevance more clearly in the results and discussion, in particular by now showing results for a plasma membrane as a highly relevant biological system, as described above. We hope that the new results for a plasma membrane membrane mimetic, with two distinct flip-flop mechanisms for two lipids of different levels of polarity, demonstrate both the richness of different flip-flop mechanisms and the power of the method. In particular, we now show that AIMMD is readily applicable also to complex and biologically relevant systems.

Response to Reviewer #2:

The manuscript by Matthias Post and Gerhard Hummer provides valuable insights into the mechanisms of lipid flip-flop and nanopore formation. The authors apply transition path sampling (TPS) within the Artificial Intelligence for Molecular Mechanisms (AIMMD) framework, recently introduced by the same research group. By comparing coarse-grained force fields (Martini 2) with all-atom simulations (CHARMM36), lipid tail length, and the role of the pore formation process, the study offers a detailed perspective on the mechanism of lipid flip-flop without the use of biasing potentials, which were commonly employed in previous studies relying on enhanced sampling methods. The topic is highly relevant and grounded in modern computational methodologies. The results are of broad interest to the community. However, the manuscript, in its current form, lacks clarity in several areas. Additionally, certain methodological aspects require further discussion and justification. Given the significance and quality of the study, I recommend publication after revisions addressing the issues outlined below.

RESPONSE: We thank the Reviewer for the very positive assessment and encouragement. In the following, we address the comments and criticisms.

Comments:

1. Page 2: The authors state: “As coordinate for the presence and size of a possibly associated pore, we use ξ_P by Hub [36].”. Is the rationale for selecting this collective variable (CV) discussed? The work by Bubnis and Grubmüller (10.1016/j.bpj.2020.10.037) is mentioned but not elaborated upon, while the work by Rivel and coworkers was omitted (10.1021/acs.jcim.4c01960)—should the authors clarify why Hub’s collective variable (CV) was chosen over others? Particularly, the alternative collective variables seem to point the influence of the lipid acyl chains in the pore formation process. How could this relate to the defect formation process assisting lipid flip-flop?

RESPONSE: We thank the Reviewer for raising these questions. In response, we discuss the different coordinates in more detail, now also citing the work of Rivel et al. We now explain that we chose the coordinate of Hub because of its narrow focus on the presence of a water pore without consideration to lipid orientation. In this way, we can more easily separate the relative mechanistic importance of poration over lipid organization.

In the Introduction we added to the AIMMD paragraph:

“In AIMMD, we apply transition path sampling (TPS) to harvest reactive trajectories without the application of bias forces or the choice of predefined collective variables or reaction coordinates. From TPS, we learn the commitment probability (or, in short, committor) on-the-fly, encoded in a deep neural network. As the probability to proceed to the product state for given a starting configuration, the committor pinpoints important microscopic features describing the reaction mechanism. The features used as inputs for the neural net include in particular the positions of neighboring lipids in a symmetry invariant form (i.e., their transversal distance between heads, Δz^{PO4} , and the distances between individual pairs of atoms, Δr^{all} , as depicted in Fig. 1b). In addition, we include reporters on the nearby hydration, with the water-pore coordinate ξ_P of Hub as a primary input. From the influence of the features on network accuracy we then deduce the importance of factors ranging from lipid orientation to water nanoporation. For the latter, we benefit from extensive earlier studies. [with reference to Marrink and Berendsen to Hub, Bubnis and Grubmüller, and Rivel et al.]”

Note that we also added a panel to Fig. 1 in response to the suggestion of Reviewer 1.

2. Page 4: Before referring to Figure 2b, it would be appropriate to introduce explicitly the different features, as well as the related models, potentially in a visual way. In the current version, the manuscript lacks clarity on these definitions, as not all the features are introduced in the text. Consequently, the reader should refer to Figure 2 caption, which itself refers to Table 1 (which is not mentioned in the main text), while looking to the details given in the Methods section where the reader is referred to Table S1. This brings unnecessary complexity. As an example, the sentence “We find that a full description of the tunnel transition mechanism requires the relative distances, $x = \Delta r^{all}$ (red), of beads of the lipid neighbor network.” is the first reference to Δr^{all} . The referenced figure does not explicitly mention $x = \Delta r^{all}$, and the color of the Nall model appears to be orange instead of red. The confusing definitions of the features and models render the manuscript rather hard to read. After providing a clear

introduction of the features and related models, it would be advisable to check for the consistency in the terminology throughout the manuscript to ensure clarity. It may be necessary to leave in the Methods only the formal description of the input features (and to refer explicitly to the given subsection in the Results section, to guide the reader).

RESPONSE: We thank the Reviewer for alerting us of the lack in clarity in our figure and text. In response, we introduced the input features of the neuronal network in the introduction (see preceding response). We also added a third panel to Fig. 1 for visual aid. We consistently changed the x-labels of Fig. 2b.

3. Page 4: The authors underlined that: “This implies a simple shape of the committor, $\Phi \approx \phi(x \cdot v)$, i.e., a linear model of the input features.”. Could the authors elaborate on this assumption in relation to Figure 2, particularly the isolines in subplot e and/or subplots c and d, as this may reinforce their statement?

RESPONSE: We deduced this surprising result from the analysis in Supplementary Fig. 2, where the direction of the gradient clearly does not change much as one moves from lipids strongly committed to the upper leaflet ($\phi \approx 0$) via the transition state ($\phi \approx 1/2$) to lipids committed to the lower leaflet ($\phi \approx 1$). When using this new linear coordinate, $x \cdot v$, then as a single input feature for the committor model (thereby enforcing a form $\phi(x) = \phi(x \cdot v)$, we found that the resulting quasi-linear model had the same prediction accuracy as one including all coordinates individually as input (Supplementary Figure 2c) comparing $\phi(x)$ from the fully nonlinear neural net to $\phi(r(x) = x \cdot v)$ as a 1D nonlinear function of a linear projection, $r(x) = x \cdot v$. The committor isolines shown in the panels show some projection artifacts due to the averaging. Also, in Fig. 2, due to the boundary condition, the lines bend at the states, but only where there is no data. We added this to the latter discussion:

“The projection $\Delta r^{\text{all}} \cdot v_r$ of distances onto v_r resolves the configurations x according to their committor values ϕ (Fig. 2d, and the (gray) iso-lines of $\phi(x \cdot v)$ in Fig. 2e). Note, though, that close to the state boundaries, defining the $\phi = 0$ and 1 iso-surfaces, this simple linear model has to fail. For a given configuration, the linear projection $\Delta r^{\text{all}} \cdot v_r$ identifies the features that commit it to one or the other leaflet as the probe head and leaflet phosphates approaching each other (Panel V and panel VI).”

4. Page 4: The authors wrote: “Yet, most of the weight is found in the distances of the probe head to neighboring lipids (Fig. S2).”. It is unclear how this is inferred from Figure S2.

RESPONSE: We revised Supplementary Fig. 2 by adding a panel that makes the relative weights of the distances clearer.

5. Page 5: The authors estimate nanopore persistence time as $\sim 0.4 \mu\text{s}$ based on Figure S6a. The methodology relies on counting closure events in 200 ns trajectories. Could the authors discuss whether this method provides a maximum likelihood estimate? Additionally, how does this compare with estimates and methodology in Bennett et al. (<https://doi.org/10.1016/j.bpj.2013.11.4486>, e.g. Fig. 5)? The $0.4 \mu\text{s}$ estimate is also cited on pages 7 and 8—these references may need revision.

RESPONSE: In response, we added a paragraph to the Methods: “**Water pore lifetime.** Starting from shooting point structures of the last MC step of the DMPC pore-mediated flip-flop, we started free, unbiased simulation runs, using the same parameters as before, for either a maximum of 200 ns or until collapse of the pore was observed. We then calculated the mean pore lifetime using a maximum-likelihood estimate for randomly censored data with exponential kinetics, $\sum_{i=1}^N t_i/n$, where t_i is the duration the pore stayed open in simulation run i , either before closing spontaneously in n of the N runs or before the run was terminated with still intact pore in the remaining $N - n$ runs. One recognizes in the estimator the ratio of the aggregate time of being uninterrupted in the open state divided by the number of closing transitions. This estimator maximizes the likelihood $L(k) = \prod_{i=1}^n p(t_i|k) \prod_{i=n+1}^N S(t_i|k)$ written as a product of the survival probability $S(t|k) = e^{-kt}$ (for terminated runs) and the corresponding probability density $p(t|k) = -dS/dt = ke^{-kt}$ (for runs in which the pore closed). We also changed the title of the section in the main text to “Pore nucleation precedes flip-flops.”

6. Page 6: The authors attribute the lower accuracy of the atomistic model compared to the coarse-

grained model to sample size. Does this also explain the larger dispersion in the accuracy reported on Figure 3 (compared to that of Figure 2)?

RESPONSE: We thank the Reviewer for pointing this out. In response, we now explain in the text: “The somewhat lower accuracy of the committor models for the atomistic DMPC model (α between 0.8 and 0.9 in Fig. 3) compared to the Martini DMPC model ($\alpha \approx 0.9$ in Fig. 2b) is likely due to a combination of a more complex mechanism in the atomistic simulations (with pore and tunnel mechanism) and fewer training data.” To address the issue of sample size, we now show in Supplementary Fig. 7a how the accuracy parameter estimates depend on sample size for artificial data. We find that the median of the accuracy parameter α is robustly estimated, largely independent of sample size, but the scatter decreases substantially as the sample size is reduced.

7. Page 9: The statement “The middle-ground might thus be what we observed for atomistic DSPC lipids, where local membrane thinning combined with narrow water threads to establish a passageway for lipid flip-flop.”. However, earlier results suggest that in DMPC, flip-flops are rare during pore nucleation and do not occur in the presence of narrow water threads. Does this truly represent a middle ground?

RESPONSE: We apologize for the confusion caused by unclear statements. In light also of the new results, we changed the sentence to: “A middle-ground between a completely dry tunnelling and nanopore formation might thus be what we observed for atomistic DSPC lipids and for PLPC lipids in the plasma membrane, where the rare local membrane thinning combined with narrow water threads and nanodroplets to establish a passageway for an even rarer lipid flip-flop.”

8. Page 10: “Pathway Π_P required the preparation of a water pore by introduction of a flat-bottomed position restraint of the lipids in the center of the simulation box.”. The authors chose to use Hub’s CV as a measure of the pore state but decided to use inverted flat-bottom potentials to create the membrane pore, which is unlikely to follow a minimal free energy path (MFEP) during the pore formation process in the membrane. Could the authors comment on this choice?

RESPONSE: We thank the Reviewer for raising this important point. In response, we added the following text to the discussion: “As strong evidence for the dominance of the pore pathway Π_P here, we first improved the statistics by running multiple TPS MC chains starting from different seed paths that jointly cover the two extreme mechanisms of a pre-existing pore and of dry lipid tunneling. Importantly, already the first paths in each chain were unbiased transition trajectories, albeit from a transition state (here, with a lipid at the bilayer center or with a pore) created by gently applying restraints. As new transition paths were discovered, memory of the seed paths was quickly lost (Supplementary Figure 6c). We even observed that the character of the transition state changed: in all runs starting with dry tunnelling Π_T , water pores formed eventually, leading to a Π_P mechanism (Fig. 3a). This pathway via nanopores then persisted for all TPS walkers, ensuring the convergence to the unbiased, equilibrium TPE of our atomistic DMPC membrane.”

9. Page 10: “To this end, we performed 1 ns (10 ns in case of Martini) of simulations with $k = 500$ kJ mol⁻¹nm⁻² and r ranging from 0.5 (head) to 1.6 nm (tail) to open the pore.”. There is a lack of details to reproduce this procedure. Particularly, it seems that the change in the radius of the flat portion of the potential is not increased continuously; however, it is not mentioned how many steps were necessary to reach a 1.6 nm radius. The authors do not show that the system had time to relax after increasing the pore radius, neither between two steps used to increase the pore radius, nor after the last one (which is 10 times longer than intermediate steps) that is used to produce the shooting points for the TPS procedure. Using CV designed to follow the MFEP of pore formation, both Hub et al. and Rivel et al. point out that at least longer simulation times are necessary (> 100 ns) for AA simulations at a given pore size to obtain a satisfying sampling for an umbrella sampling procedure.

RESPONSE: Here we refer the Reviewer to the preceding response. In a nutshell, our transition path sampling starts from an initial configuration, which we prepared to have specific characteristics by gently perturbing equilibrium configurations (here: by placing a lipid in the bilayer center and by opening a pore as the two extreme transition states). We then let the system evolve in TPS according to completely unbiased dynamics, i.e., with regular MD runs. As new transition paths are discovered, the

system relaxes and memory of the seed path is lost. In our case, an initial pore relaxes quickly to the state typical of the respective transition state, as shown by the blue lines in Figs. 2a and 3a, where the pore size relaxes to a typical value by opening or contracting, with the variance indicative of variations in the pore size at this particular transition state. Over longer sampling periods, some transition path samplers even transition from one “reaction channel” (say, with a dry transition state) to another (say, with a pore-like transition state).

10. Page 10: "We used the last 1 ns (10 ns for Martini) to pool shooting points for parallel TPS using AIMMD (see below)". Were the shooting points selected randomly? At an even step? Is that enough to avoid any memory effect between frames?

RESPONSE: The shooting points were chosen randomly, as it is done by the AIMMD algorithm and described in detail in Jung et al. (2023). AIMMD biases the shooting point selection to points near the transition state (with committor values of 1/2), but corrects for this bias in the usual way of a Metropolis Monte Carlo sampling scheme (“We allow for some deviations of shooting from the optimal $\phi = 0.5$ iso-surface by sampling from a Cauchy distribution of the logit q of ϕ ($q = \ln \left[\frac{\phi}{1-\phi} \right] \sim \text{Cauchy}(\mu=0, \gamma=1)$).”) Jung et al. (2023) reported rapid decorrelation (after about 4 paths). Here, a visual inspection of the series of pore coordinate values in Figures 2a and 3a indicate a similarly rapid decorrelation locally for a given transition state. However, a complete change in mechanism with a substantially different mechanism takes longer, about 50 steps for the transition state to change from nanopore to nanowire (Fig. 2a), and about 500 steps to transition between dry and wet transition states in Martini and atomistic MD simulations (Fig. 2a and 3a). The latter transition can be thought of as an activated transition in the space of trajectories, leading from one predominant saddle to another saddle of lower free energy.

11. Page 10: The authors mention that "[p]reparation of initial Π_T trajectories was achieved by a harmonic constraint pulling the probe lipid headgroup with $v = 0.001$ nm/ps, $k = 1000$ kJ mol⁻¹ nm⁻² [...]", while for pathway Π_P , the authors mention that they "performed 10 ns (100 ns in Martini MD) of simulations, in which [they] also fixed the probe lipid in the middle of the bilayer using an additional cylindrical harmonic restraint of the PO4 group with $r = 2$ nm, $k = 1000$ kJ mol⁻¹ nm⁻²". Two different strategies seem to have been used to bring the probe lipid close to the tipping point. A steered MD procedure is detailed concerning the preparation of initial Π_T trajectories, but the preparation of Π_P is not as detailed. Could the authors add a description of the latter and comment on the difference between the procedures?

RESPONSE: We hope that with the changes to the Methods, the section can now be better understood. The goal is to create initial trajectories for both mechanisms. In Π_T , this can roughly be achieved by pulling, the main challenge in Π_P is to open the pore and not the flipping and sampling from the restrained trajectory was expected to quickly result in accepted TPs.

12. Page 11: The authors state that they used "MDAnalysis to define the two final states of the transition by the transversal displacement z from the midplane (defined by the lipid P atoms (PO4 bead for Martini) w.r.t. the vertical center ($z = 0$) of all P's), namely the state L when $z < -1.3$ nm, and state U when $z > 1.3$ nm, respectively". However, they don't comment on the motivation for the choice of such an arbitrary threshold. Particularly, how using a unique threshold regardless of the studied lipid type (DMPC v. DSPC) can influence the results?

RESPONSE: We used the headgroup density along z of the initial unbiased simulations for evaluating what threshold to use. We added: "... with respect to the vertical center ($z = 0$) of all P's). Based on the distribution of heads in the initial equilibrium simulations, we set the state \mathcal{L} when $z < -1.3$ nm ... (1.7 nm for DSPC, 1.65 nm for cholesterol and 1.9 nm and PLPC of the plasma membrane)".

Minor comments / typo:

1. Page 3: (minor/typo) The phrase “referring to Methods for details of the TPS setup” is vague given the number of subsections in the Methods. It would improve readability to guide the reader more specifically.

RESPONSE: We now write: “referring to Methods for detailed descriptions of the MD simulations and TPS setup”

2. Page 4: (minor/typo) The sentence “To further explain how they instead tunnel through the bilayer, we study the importance of individual microscopic features describing the committor $\Phi(x)$.” may read more fluidly if rephrased e.g.: “[...] of individual microscopic features x in describing the committor $\Phi(x)$.”

RESPONSE: We changed this accordingly.

3. Page 4: (minor/typo) The term “transition times” is introduced in “Notably, in this intermediate, unstable period, the corresponding transition times are the smallest, even compared to ΠT (see Fig. S1c in the SI).” However, at this point in the manuscript, it is unclear how transition times are defined. It would be recommended to define this term, or to refer where we could find its definition.

RESPONSE: We added “transition times to move from L to U (or vice versa)”.

4. Pages 3-4: (minor/typo) The manuscript states, “The flip-flop transition of an individual lipid (the ‘probe lipid’) is tracked by its transversal displacement z from residing in the lower leaflet (state L) to the upper leaflet (state U). Here, we time-order the TPs as $L \rightarrow U$.” However, Figure S1a referred in the following paragraph illustrates a flopping event ($U \rightarrow L$). I am not sure whether it is necessary to indicate that the TPs are time-ordered, as I did not see where this information was relevant later in the manuscript. On the other hand, this may bring confusion to the reader.

RESPONSE: We removed this sentence, as opposed to changing all Supplementary figures.

5. Page 4: (minor/typo) The manuscript states, “After about $n \approx 500$ MC steps, we have a behavior similar to ΠT , and thus, all water beads are flushed out and the pore is closed completely during the remaining transitions.”. However, in Figure 2a, the closure appears to occur closer to 600 or 700 MC steps. Please verify.

RESPONSE: We admit that with the same colors, it might be hard to see that 4 of the 6 samplers have already made the transition before the 500 steps. The actual characteristic number of MC steps required to transition to the new mechanism is thus likely to be less. Similarly, in the atomistic simulations 50% of the samplers have transitioned from a dry to a wet mechanism (Fig. 3a) in about 300 MC steps. Nevertheless, we kept the more conservative estimate of ~ 500 MC steps in the text.

6. Page 4: (minor/typo) The authors refer to “ $\Delta r_{\text{all}} \cdot v_r$ ” as the “mean neighbor distance,” while it seems to be the projection of this quantity on v_r .

RESPONSE: Thank you, we changed it to “projected neighbor distance” because not all v entries are positive weights.

7. Page 4: (minor/typo) “E.g. a conformation with joined tails [...]”. It may be advisable to avoid beginning the sentence with “E.g.”

RESPONSE: Thank you, we changed this to a colon.

8. Page 6: (minor/typo) “If we do, see Figure 3c, we see that again the committor prediction using distances Δr_{all} to all neighboring lipid atoms (yellow) outperforms a simple model using z alone (light blue).” Reconsider the phrasing; some commas may be missing.

RESPONSE: We instead wrote “we see that the committor prediction using all neighbor distances Δr^{all} (yellow) again outperforms a simple model using z alone (light blue)”

9. Page 9: (minor/typo) Indicate the unit when discussing salt concentrations.

RESPONSE: We added the units.

10. Page 9: (minor/typo) "after which we performeda" → add a space.

RESPONSE: Done.

11. Page 9: (minor/typo) The statistical ensemble used in the short equilibration run is not clearly defined. Please provide additional details.

RESPONSE: We thank the Reviewer to point out this mistake, which we now corrected by adding the word "both".

12. Page 10: (minor/typo) "and r ranging from 0.5 (head) to 1.6 nm (tail) to" → define explicitly what "r" represents.

We added "distance to the center."

13. Page 10: (minor/typo) "See Fig. Fig. S12e for snapshots of one of these initial" → Typo.

RESPONSE: Done.

14. Page 10: (minor/typo) The authors provide the equation of the loss function used in the training of the neural network involved in the AIMMD framework. It seems that the sum from $i=1$ to N refers to the N features. That is not defined explicitly. Meanwhile, in the same paragraph, the authors indicate that they produce " $N = 1000$ MC steps for ΠP ".

RESPONSE: We rewrote the sentence with the equation to make it clearer.

15. Page 10: (minor/typo) "To accelerate the learning of the committor, we include the SPs of the initial restraint runs to the training set." The abbreviation "SP" for shooting point was not defined earlier.

RESPONSE: We thank the Reviewer for finding this inconsistency and included the abbreviation from the start.

16. Page 11: (minor/typo) "w.r.t." → It may be recommended to not use such an abbreviation.

RESPONSE: We followed that recommendation.

17. SI (minor/typo) In the legend of Fig. S1 (and some others) in the SI, the authors mention that the subfigure "a" represents an "exemplary trajectory." The authors may consider rewording it or explaining in what manner this trajectory was selected to be particularly representative.

RESPONSE: We added a few explanatory words to each of the Supplementary illustrations.

18. SI (minor/typo) In the legend of Fig. S1: "k-neighrest neighbor" → "k-nearest neighbor."

RESPONSE: Done.

Reviewer #3 (Remarks to the Author):

The authors present a detailed mechanistic study of lipid flip flop transitions in both atomistic and coarse-grained forcefields for DMPC and DSPC lipid membranes. In particular, the use of an AI-guided transition path sampling approach provides detailed mechanistic description at the atomistic scale of the order of events along the lipid flip flop transition pathway. The application of transition pathway analysis of this system, particularly at the atomistic scale, would be of interest to the biomolecular simulation and rare events community. As pointed out by the authors, these results may hint at a more general linearization effect in high dimensional feature spaces, although this is not conclusively established in this work.

I have a few questions for the authors to potentially improve the manuscript.

RESPONSE: We thank the Reviewer for the positive feedback and constructive suggestions.

1. How well are the neural network model weights fitted? I recommend adding a supplementary figure demonstrating convergence of the NN weights as measured by minimization of the loss function over training epochs, or some suitably related metric.

RESPONSE: We added Supplementary Fig. S16, as mentioned in the main text. Panel (a) shows that the loss converged to a reasonable number, where a value of 0.6 roughly corresponds to good predictions for close-to-optimal probabilities to generate reactive transition paths. Panel (b) then shows the loss while training neuronal network models post-simulation, highlighted in Fig. 2b, as a function of number of epochs.

2. For the Charmm36 DMPC model described in Figure 3, what is the likelihood of a reverse transition of the TPS process back to the dry transition tube? It seems clear from this data that the dry pathway is a locally metastable transition tube which eventually transitions to the wet transition tube via the TPS process. However, the average number of MC steps required to make this transition from the dry to the wet tube seems on average much longer than the wet-initialized tubes MC processes were simulated. If these were extended, would the reverse process be observed and with what relative probability? Since the global stability of the wet transition process is a key result of this paper, I would suggest extending the TPS sampling or otherwise quantifying the stability of the wet process for the atomistic case.

RESPONSE: The Reviewer raises an interesting point concerning the relative reactive flux carried by the two reaction mechanism, tunnel and pore. Following the suggestion of the Reviewer, we now roughly doubled the TPS chains starting from the pore mechanism (Fig. 3a,b). However, we still did not observe a single switch from a pore to a tunnel mechanism. With roughly equal aggregate numbers of TPS Monte Carlo steps in either mechanism, and 8 transitions observed from tunnel to pore and none in reverse, one can roughly expect that the reactive flux carried by the tunnel mechanism is at least 8 times lower than that carried by the pore mechanism. While this statistical analysis could be worked out in more detail by assuming Poisson processes for mechanism switch, we believe that the numbers are sufficiently clear.

We note further that we already gave an argument in the manuscript, comparing the flip-flop transition times with those of the much longer pore life-time.

3. Further, showing that one transition tube is meta-stable and the other is not meta-stable only qualitatively supports the conclusions, but I would suggest that the relative path probabilities of one tube over the other would more precisely quantify and support this point. A similar argument could be suggested for quantifying the relative probabilities of the order of molecular events sampled in the atomistic DMPC case. Since this evidence is central to the conclusions of the paper, it may be worth adding this analysis to strengthen the conclusion.

RESPONSE: As discussed in the last point, one would need to sample the transition rate back into the other mechanism to estimate the true relative path probabilities. As we already did extend the MC

simulations and no back-transitions were found, we do not think that this is really feasible or necessary to do (in the context of this work).

Minor suggestions to improve the presentation:

1. For me, it would help to enlarge or rearrange the text placed inside figure panels and on the axis markings in figures 2,3 and 4. I had a hard time reading them at first.

RESPONSE: We agree that the text might be a little small but with the font size we follow the suggestions of the Nature figure style guide.

2. Although an extensive presentation of AIMMD is beyond the scope here, a short summary of the AIMMD method would be helpful.

RESPONSE: Following this suggestion, we added some additional explanation to the AIMMD description in the Methods section.

3. I believe the methods details for the DPSC model is missing and should be added.

RESPONSE: We thank the Reviewer for pointing out this omission. We added the differences in the setup. We note that we also added the simulation of a model plasma membrane.

2nd Revision

RESPONSE TO REVIEWER COMMENTS

Response to Reviewer #1:

During revision, the manuscript has substantially improved. The methodology presented in the paper is an excellent addition to the toolkit and supports the acceptance of the manuscript.

RESPONSE: We thank the Reviewer for the helpful response to the revised document. In the following, we address their remaining critical remarks.

There remains some degree of discrepancy between the atomic-level simulation results and the results of coarse-grained models (DMPC). In this respect, the revised paper clearly raises the issue and justifies the discussion of the coarse-grained model results by stating that it can be used to demonstrate an AI-based analysis of the "dry" mechanism. It is also true, just as the paper states, that there is no convincing experimental evidence that lipid flip-flop occurs in DMPC membranes through pores, or without them.

RESPONSE: We are glad that this point is now clear.

However, the authors should consider addressing one more theme in their paper. Experimentally, first of all, it is known that the means used by cells to protect their energy reserves, which are based on differences in salt ion concentrations across the plasma membrane, is cholesterol: when the membrane cholesterol concentration is increased, the formation of pores can be prevented (as confirmed by electroporation experiments). On the other hand, it is known that without cholesterol, cell membranes are very elastic (low bending rigidity), while at high cholesterol concentrations, cell membranes become more rigid (high bending rigidity). From this experimental observation, one could indirectly conclude, or at least make an educated guess, that in elastic cell membranes, lipid flip-flop is mainly based on pores, while in rigid cell membranes, lipid flip-flop does not require the presence of pores. This may also partly explain the MARTINI model results described in the paper, considering that the ability of the MARTINI model to describe the elasticity of membrane structures is not quantitatively very convincing.

RESPONSE: We thank the Reviewer for this very informed remark. While we did not explicitly analyze the bending rigidity of the bilayers used in this work, it is fair to speculate that membrane flexibility has an influence on the flip-flop mechanism. As the Reviewer stated, other studies—of which some were already represented by citations in our work—indicate that a lower bending rigidity results in a lower free energy barrier to pore formation, opening up this particular pathway for lipids to flip.

As shortly mentioned in the manuscript, we see that the thicker DSPC and plasma-membrane membrane do locally thin before the flip-flop occurs, indicating that the bending modulus might play a role even if pores do not fully form.

We added to the discussion:

“Also, a higher membrane bending rigidity (e.g., due to cholesterol) should suppress both pore formation and lipid flip-flop, as suggested here by the pronounced local bulging and thinning of the DSPC and plasma membranes, compared to the pore-forming DMPC bilayer.”

However, we do not think the different mechanism of the Martini3 model should be contributed to bending rigidity (which is also claimed to be close to all-atom, <https://doi.org/10.1021/acscentsci.5c00755>), but to the artificial behavior of the water beads. We added as a remark in the discussion:

“... may flip without water (at least in part due to the well-known instability of Martini water pores)”

Response to Reviewer #2:

The authors have carefully addressed all my comments and remarks. Moreover, they extended the work to a realistic membrane model, which represents a substantial and valuable addition to the original manuscript.

The article was already of great importance and well-founded in its initial version. After the modifications, the manuscript is easier to follow, and the message is both clearer and stronger.

The authors should be commended for the quality of this work and the thoroughness of their revisions. I fully recommend the acceptance of this article in its current form.

RESPONSE: We thank the Reviewer for this very kind feedback and the positive assessment.

Reviewer #3 (Remarks to the Author):

The authors present a detailed mechanistic study of lipid flip flop transitions in both atomistic and coarse-grained forcefields for DMPC lipid membrane, DSPC lipid membrane and a plasma membrane mimetic. In particular, the use of an AI-guided transition path sampling approach provides detailed mechanistic description at the atomistic scale of the order of events along the lipid flip flop transition pathway. The additions the authors have made in the revised version have significantly strengthened the manuscript I would recommend publication in it's current state.

RESPONSE: We appreciate the Reviewer's kind evaluation and the positive assessment.